# Mitosis can drive cell cannibalism through entosis

Joanne Durgan[1,2], Yun-Yu Tseng[2,3], Jens C Hamann[2,4], Marie-Charlotte Domart[5], Lucy Collinson[5], Alan Hall[2†], Michael Overholtzer[2], Oliver Florey[1*]

[1]The Babraham Institute, Cambridge, United Kingdom; [2]Memorial Sloan Kettering Cancer Center, New York, United States; [3]Weill Graduate School of Medical Sciences, Cornell University, New York, United States; [4]Louis V Gerstner Jr Graduate School of Biomedical Sciences, New York, United States; [5]The Francis Crick Institute, London, United Kingdom

**Abstract** Entosis is a form of epithelial cell cannibalism that is prevalent in human cancer, typically triggered by loss of matrix adhesion. Here, we report an alternative mechanism for entosis in human epithelial cells, driven by mitosis. Mitotic entosis is regulated by Cdc42, which controls mitotic morphology. Cdc42 depletion enhances mitotic deadhesion and rounding, and these biophysical changes, which depend on RhoA activation and are phenocopied by Rap1 inhibition, permit subsequent entosis. Mitotic entosis occurs constitutively in some human cancer cell lines and mitotic index correlates with cell cannibalism in primary human breast tumours. Adherent, wild-type cells can act efficiently as entotic hosts, suggesting that normal epithelia may engulf and kill aberrantly dividing neighbours. Finally, we report that Paclitaxel/taxol promotes mitotic rounding and subsequent entosis, revealing an unconventional activity of this drug. Together, our data uncover an intriguing link between cell division and cannibalism, of significance to both cancer and chemotherapy.

**\*For correspondence:** oliver. florey@babraham.ac.uk

†Deceased

**Competing interests:** The authors declare that no competing interests exist.

## Introduction

Cellular cannibalism is an ancient form of feeding used by bacteria (*González-Pastor et al., 2003*) and predatory amoebae (*Waddell and Duffy, 1986*) in response to starvation. A similar phenomenon is observed among epithelial cells in human cancer (*Brouwer et al., 1984*; *Overholtzer et al., 2007*; *Overholtzer and Brugge, 2008*; *Sharma and Dey, 2011*; *Yang and Li, 2012*; *Cano et al., 2012*), suggesting this primeval process may promote survival within the tumour microenvironment (*Fais, 2007*; *Matarrese et al., 2008*; *He et al., 2013*; *Lozupone and Fais, 2015*). Homotypic epithelial cell cannibalism can occur by entosis, an intriguing process through which one live and viable cell is completely engulfed by another, yielding a 'cell-in-cell' structure (*Overholtzer et al., 2007*). The vast majority of internalised, entotic cells are ultimately killed and digested by their hosts, through a mechanism involving non-canonical autophagy and lysosomal degradation (*Yuan and Kroemer, 2010*; *Florey et al., 2011*).

Entosis is observed in a wide range of human cancers (*Overholtzer et al., 2007*; *Overholtzer and Brugge, 2008*) and is believed to mediate pleiotropic effects on cancer biology. On one hand, entotic cell killing can limit outgrowth through the elimination of internalised cells (*Florey et al., 2011*), representing a possible means of tumour suppression. Conversely, entosis simultaneously promotes host cell survival and transformation, by providing valuable nutrients (*Fais, 2007*; *Krajcovic et al., 2013*) and driving genomic instability (*Krajcovic et al., 2011*). Consistent with these pro-tumorigenic effects, the frequency of entosis is found to increase with tumour grade (*Krajcovic et al., 2011*), and cell-in-cell formation correlates with poor patient outcome

**eLife digest** For over a century, scientists looking down microscopes at samples from human cancers have noticed cells eating other cells – in other words, cell cannibalism. The causes and mechanisms involved in this unusual process, which is also known as entosis, are not well understood and its relationship to cancer is complex. On one hand, cell cannibalism may promote cancer by providing nutrients for growing tumours and making it more likely that genetic errors will occur. On the other hand, this process may resist cancer by eliminating damaged cells.

In the laboratory, cell cannibalism has only been seen in cells that are detached from their surroundings. Cells in the body are typically surrounded and supported by a mesh of proteins called the extracellular matrix. However, within a tumour, cancer cells can often begin to grow without being attached to the matrix, which means that cell cannibalism can occur. A protein called Cdc42 plays a part in how cells attach to each other and to the extracellular matrix, but the role of Cdc42 in controlling entosis had not been previously explored.

Durgan et al. initially set out to ask whether Cdc42 was involved in the established process of cell cannibalism, as seen in detached cells. However, the experiments showed that removing Cdc42 from human cells grown in the laboratory had little effect on this method of entosis. Unexpectedly, though, the loss of Cdc42 did enable a different form of cell cannibalism in cells that remained attached to the extracellular matrix, which had not been seen before. This new cannibalism process is linked to cell division, with cells that are dividing or that have recently divided being consumed by neighbours. This form of cell cannibalism is more commonly seen in cancers where the cells divide a lot, and some chemotherapy drugs that interfere with cell division also increase the rate of cell cannibalism.

During cell division a group of proteins – including RhoA and myosin – cause cells to become rounder and stiffer. Durgan et al. suggest this allows the dividing cells to force their way inside other cells, the key first stage of entosis. Since cancer cells divide often, this form of cell cannibalism may lead to the cancer cells being destroyed by their healthy neighbours, in a form of "assisted suicide". This reveals an unexpected link between cell division and cell cannibalism, which is relevant to both cancer and chemotherapy. Future work will explore whether entosis can be used to predict how a cancer will progress in a patient, or how they will respond to a given treatment.

---

(*Schwegler et al., 2015*; *Schenker et al., 2017*), suggesting this process may be associated with tumour progression. Finally, entosis can mediate cancer cell competition (*Sun et al., 2014a*), allowing one population to preferentially engulf and kill another, and may therefore contribute to shaping tumour evolution. Together, these findings indicate that entosis can mediate both tumour suppressive and promoting effects (*Krishna and Overholtzer, 2016*), but the overall impact of entotic cell cannibalism on tumour biology and progression remains to be fully understood (*Durgan and Florey, 2015*).

Mechanistically, entosis involves the formation of adherens junctions and the generation of acto-myosin-based contractility, which enables one cell to actively push or 'invade' into a more deformable neighbour, in an unconventional mode of engulfment (*Overholtzer et al., 2007*). This process is known to be triggered by matrix deadhesion, which renders the cells unanchored and this contractile force unopposed. ROCK-mediated myosin phosphorylation is indispensable for entosis in cultured cells (*Overholtzer et al., 2007*; *Sun et al., 2014a*; *Wan et al., 2012*; *Sun et al., 2014b*) and during embryonic implantation (*Li et al., 2015*). Accordingly, regulated changes in actomyosin contractility, as induced by oncogenic K-Ras, can modulate entosis in suspension (*Sun et al., 2014a*); a similar mechanism operates during the early stages of matrix adhesion (*Wan et al., 2012*).

Rho-family small GTPases regulate many fundamental cellular processes (*Jaffe and Hall, 2005*; *Heasman and Ridley, 2008*), including entosis. RhoA controls actomyosin contractility through ROCK, and blebbing through mDia, and is therefore indispensable for cell-in-cell formation (*Overholtzer et al., 2007*; *Yamada and Nelson, 2007*; *Purvanov et al., 2014*). Similarly, Rac1 can regulate myosin phosphorylation to modulate entotic cell competition (*Sun et al., 2014a*). Other Rho-family members seem likely to influence cell cannibalism, through effects on the actomyosin

cytoskeleton, cell-cell contacts or cell-matrix adhesion, but their contributions have yet to be investigated. The present study was initiated to explore a possible role for Cdc42, a master regulator of epithelial cell biology (*Heasman and Ridley, 2008*; *Joberty et al., 2000*; *Etienne-Manneville, 2004*; *Martin-Belmonte et al., 2007*; *Jaffe et al., 2008*; *Wallace et al., 2010*; *Roignot et al., 2013*). Unexpectedly, this work has uncovered a novel mechanism of entotic cell-in-cell formation, driven by mitosis. We present our findings on the relationship between epithelial cell division and cannibalism and demonstrate its relevance to both human cancer and chemotherapy.

## Results

### Cdc42 controls cell-in-cell formation in adherent epithelial cells

To assess a possible role in entosis, Cdc42 was depleted in 16HBE human bronchial epithelial cells, using multiple distinct and non-overlapping RNAi reagents. All Cdc42-specific duplexes and hairpins yield a robust knockdown (*Figure 1a*), and induce expected functional changes, such as disruption of adherens (AJ) and tight junction (TJ) maturation (*Figure 1b*) (*Wallace et al., 2010*). To initiate entosis, cells were cultured in suspension for 8 hr (*Figure 1c–d*), inducing cell-in-cell formation among ~10% of control cells, consistent with previous reports (*Krajcovic et al., 2011*). Surprisingly, despite clear effects on AJ maturation, Cdc42 depletion has no major impact on cell-in-cell formation under these conditions, suggesting that primordial junctions are sufficient to support entosis. Strikingly, however, we found instead that Cdc42 depletion promotes robust cell-in-cell formation in adherent culture (*Figure 1e–f*), in which entosis would not be expected to occur. This surprising phenotype is reproducible and statistically significant across all RNAi reagents tested, corresponding closely with knockdown efficiency (compare siCdc42.1/2), consistent with a specific, on-target effect of Cdc42. Furthermore, adherent cell-in-cell formation is also observed in Cdc42-depleted MCF7 breast epithelia (*Figure 1g–i*), indicating that this process is consistent across multiple cell lines, derived from different tissues of origin. Together, these data reveal that Cdc42 does not play a significant role in conventional entosis, but unexpectedly controls a novel form of cell-in-cell formation among adherent cells.

### Adherent cell-in-cell formation shares the mechanistic features of entosis

The adherent cell-in-cell structures observed upon Cdc42 depletion morphologically resemble those formed through entosis. However, entosis is typically triggered by matrix detachment, occuring in suspension (*Overholtzer et al., 2007*), or during the early stages of adhesion (6–8 hr after plating) (*Wan et al., 2012*). To determine whether adherent cell-in-cell formation otherwise bears the hallmarks of entosis, additional characteristics were analysed. Firstly, Cdc42 was co-depleted with α-catenin, a core junctional component that is indispensible for entosis (*Wang et al., 2015*) (*Figure 1j–l*). α-catenin depletion yields profound AJ defects, and a corresponding reduction in cell-in-cell formation, in line with an entotic mechanism. Next, the involvement of actomyosin contractility was assessed, which drives suspension entosis downstream of RhoA/ROCK (*Overholtzer et al., 2007*). Inhibition of ROCK dramatically suppresses cell-in-cell formation among Cdc42-depleted cells (*Figure 1l*), like entosis under detached (*Overholtzer et al., 2007*; *Sun et al., 2014a*, *2014b*), semi-adherent (*Wan et al., 2012*) or in vivo conditions (*Li et al., 2015*). Consistent with this, active, phospho-myosin (pS10-MLC2) is clearly enriched in the internalising cell tail (*Figure 1m*). Finally, entotic cell cannibalism characteristically involves non-canonical autophagy (*Florey et al., 2011*) and lysosomal degradation (*Overholtzer et al., 2007*; *Krajcovic et al., 2013*). Consistent with this mechanism, both LC3 (an autophagy protein) and LAMP1 (a lysosomal marker) are transiently recruited to the vacuoles of Cdc42-depleted cell-in-cell structures (*Figure 1n*). Taken together, these data reveal that entosis can indeed occur among adherent epithelial cells and suggest that a distinct mechanism must trigger cell-in-cell formation under these conditions.

### Mitosis drives entosis in adherent cells

To investigate the mechanism underlying adherent entosis, long-term timelapse imaging was used to track live cell-in-cell formation events among Cdc42-depleted cells. Representative movies and stills are shown (*Figure 2a–c*, *Videos 1–3*). During every cell-in-cell event analysed, the inner cell

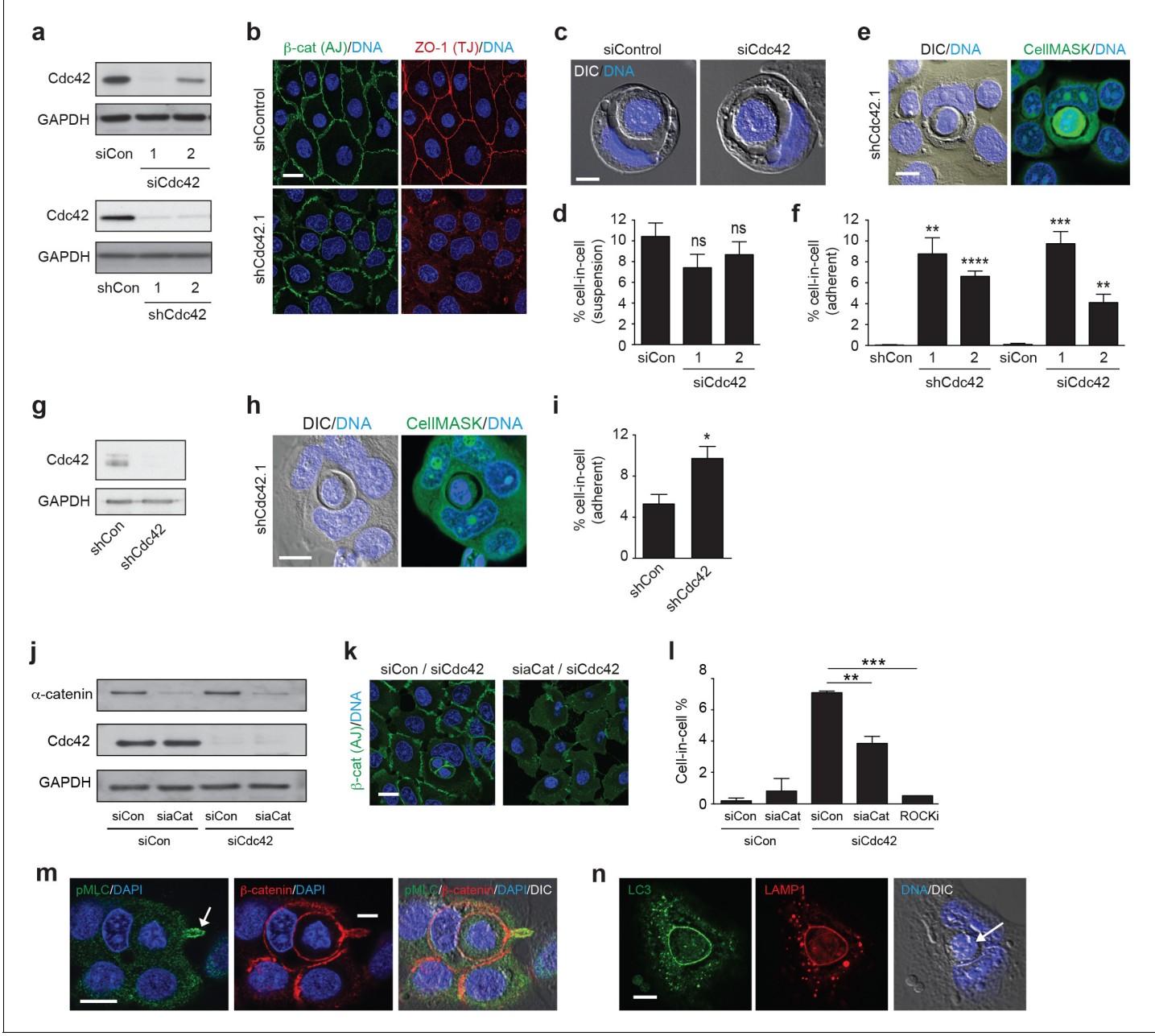

**Figure 1.** Cdc42 controls entosis in adherent epithelial cells. (**a**) Control and siRNA or shRNA Cdc42-depleted 16HBE cell lysates were probed for Cdc42 and GAPDH expression by western blotting. (**b**) Representative confocal images of control and Cdc42-depleted 16HBE monolayers stained for β-catenin (adherens junctions), ZO-1 (tight junctions) and DNA. Scale bar = 20 μm. (**c**) Representative images of cell-in-cell structures formed in matrix detached control and Cdc42-depleted 16HBE cells. Cell were stained for DNA (blue) and imaged by IF/confocal and DIC. Scale bar = 5 μm. (**d**) Quantification of suspension cell-in-cell formation in control and Cdc42-depleted cells. >200 cells were counted per sample/experiment, across three separate experiments. Error bars denote mean±SEM. ns = no significant difference, t-test. (**e**) Representative images of a cell-in-cell structure formed under adherent conditions in Cdc42-depleted 16HBE cells. Cells were stained for cell body (green) and DNA (blue) and imaged by IF/confocal and DIC. Scale bar = 10 μm. (**f**) Quantification of adherent cell-in-cell formation. >200 16HBE cells were counted per sample/experiment, across three separate experiments. Error bars denote mean±SEM. **p<0.002; ***p<0.0002; ****p<0.0001, t-test. (**g**) Control and shRNA Cdc42-depleted MCF7 cell lysates were probed for Cdc42 and GAPDH expression by western blotting. (**h**) Representative images of a cell-in-cell structure formed under adherent conditions in Cdc42-depleted MCF7 cells. Cells were stained for cell body (green) and DNA (blue) and imaged by IF/confocal and DIC. Scale bar = 10 μm. (**i**) Quantification of adherent cell-in-cell formation. >200 MCF7 cells were counted per sample/experiment, across three separate experiments. Error bars denote mean±SEM. *p<0.02, t-test. (**j**) Lysates from 16HBE cells co-depleted of Cdc42 and α-catenin (aCat) were probed for α-catenin, Cdc42 and GAPDH by western blotting. (**k**) Representative confocal images of 16HBE cells co-depleted of Cdc42 and siControl or α-catenin and stained for β-catenin (green) and DNA (blue). Scale bar = 20 μm. (**l**) Quantification of cell-in-cell structures in adherent 16HBE cells treated with siCdc42

*Figure 1 continued on next page*

*Figure 1 continued*

and siControl or siα-catenin, treated −/+10 μM Y-27632 (ROCKi), for 3 days. >200 cells were scored per sample/experiment, across three separate experiments. Error bars denote mean±SEM. **p<0.002; ***p<0.0002, t-test. (m) Confocal images of a forming cell-in-cell structure in adherent, Cdc42-depleted 16HBE cells fixed and costained for pMLC2 (S19; green), β-catenin (red) and DNA (Hoechst, blue). The arrowhead indicates the tail of the internalising cell. Scale bar = 10 μm. (n) Cell-in-cell structures in Cdc42-depleted 16HBE cells were fixed and costained for LC3 (green), LAMP1 (red) and DNA (blue), and imaged by IF/confocal and DIC. The arrowhead indicates a dying internalised cell. Scale bar = 10 μm.

The following source data is available for figure 1:

**Source data 1.**

penetrates its host either during, or shortly after, mitosis. Several permutations of this process were observed, with one or both daughters penetrating the same or different hosts, or one daughter invading the other and both entering an adherent neighbour (cell-in-cell-in-cell). To support these studies, fixed, adherent, entotic events were visualised at a mid-way point by IF/3D-correlative light-electron microscopy (CLEM; *Figure 2d*, *Videos 4–5*). In each case, mitotic cells were observed internalising into adherent neighbours. These comprehensive imaging studies establish a clear relationship between cell division and cell-in-cell formation, suggesting that mitosis may drive entosis in adherent cell populations. To test this model more directly, Cdc42-depleted cells were arrested at the G2/M boundary using a Cdk1 inhibitor (RO-3306). Strikingly, inhibition of mitosis leads to a profound decrease in cell-in-cell formation in matrix-attached, but not detached conditions (*Figure 2e*). These data indicate that cell division drives adherent, but not suspension, entosis, highlighting two mechanistically distinct routes to epithelial cell cannibalism.

## Dividing cells can be cannibalised by fully adherent, wild-type neighbours

To investigate the process of mitosis-induced entosis further, we examined the role of Cdc42 more closely, testing whether its depletion affects the internalising mitotic cell, or its host. Control and Cdc42-depleted cells were labelled red or green, respectively, and then reseeded to yield an equally mixed monolayer. Inner/outer cell identities were scored among the adherent, cell-in-cell structures formed (*Figure 2f–g*). In almost all pairs, the inner cell is Cdc42-depleted. These cells penetrate control or knockdown neighbours with equal frequency, and the results are unchanged by dye reversal (data not shown). These findings establish that loss of Cdc42 specifically promotes internalisation, enabling a dividing cell to 'invade' into an adherent neighbour, while having no detectable effect on the host. Given that wild-type cells act efficiently as hosts, we can exclude the possibility that Cdc42-depletion promotes entosis by impairing cell-matrix contacts to 'mimic' deadhesion. To verify this further, Cdc42-depleted cells were mixed with wild-type cells expressing RFP-zyxin, an adhesome component (*Horton et al., 2015*). Basal RFP-zyxin-positive foci are observed within entotic hosts (*Figure 2h*), reinforcing the conclusion that the host cell can be fully matrix-adhered. Finally, we analysed the consequences of mitosis-induced entosis under adherent conditions. Cdc42-depleted 16HBE cell-in-cell structures were followed by timelapse microscopy and the fate of the internalised cell recorded (*Figure 2i–j*). As in suspension conditions (*Overholtzer et al., 2007*; *Florey et al., 2011*), the majority of internalised cells die, by either non-apoptotic or apoptotic means, within the entotic vacuole, establishing mitotic entosis in adherent cells as a means of epithelial cell cannibalism. Together, our data demonstrate that loss of Cdc42 permits a dividing cell to 'invade' into a neighbour by entosis, and clarify that the host cell can be wild-type and fully adherent. These findings raise the intriguing concept that normal, adherent epithelia may engulf, kill and digest dividing neighbours under certain conditions.

## Cdc42 controls mitotic morphology in polarised epithelia

Our findings support a model in which epithelial Cdc42 inhibits entosis among mitotic cells. To investigate the possible mitotic functions of Cdc42, dividing cells were analysed by flow cytometry and microscopy. No gross defects in cell cycle progression are detected upon Cdc42-depletion by flow cytometric analysis of DNA content (*Figure 3a*). However, imaging studies reveal significant morphological changes during cell division. Control and Cdc42-depleted cells were visualised during the

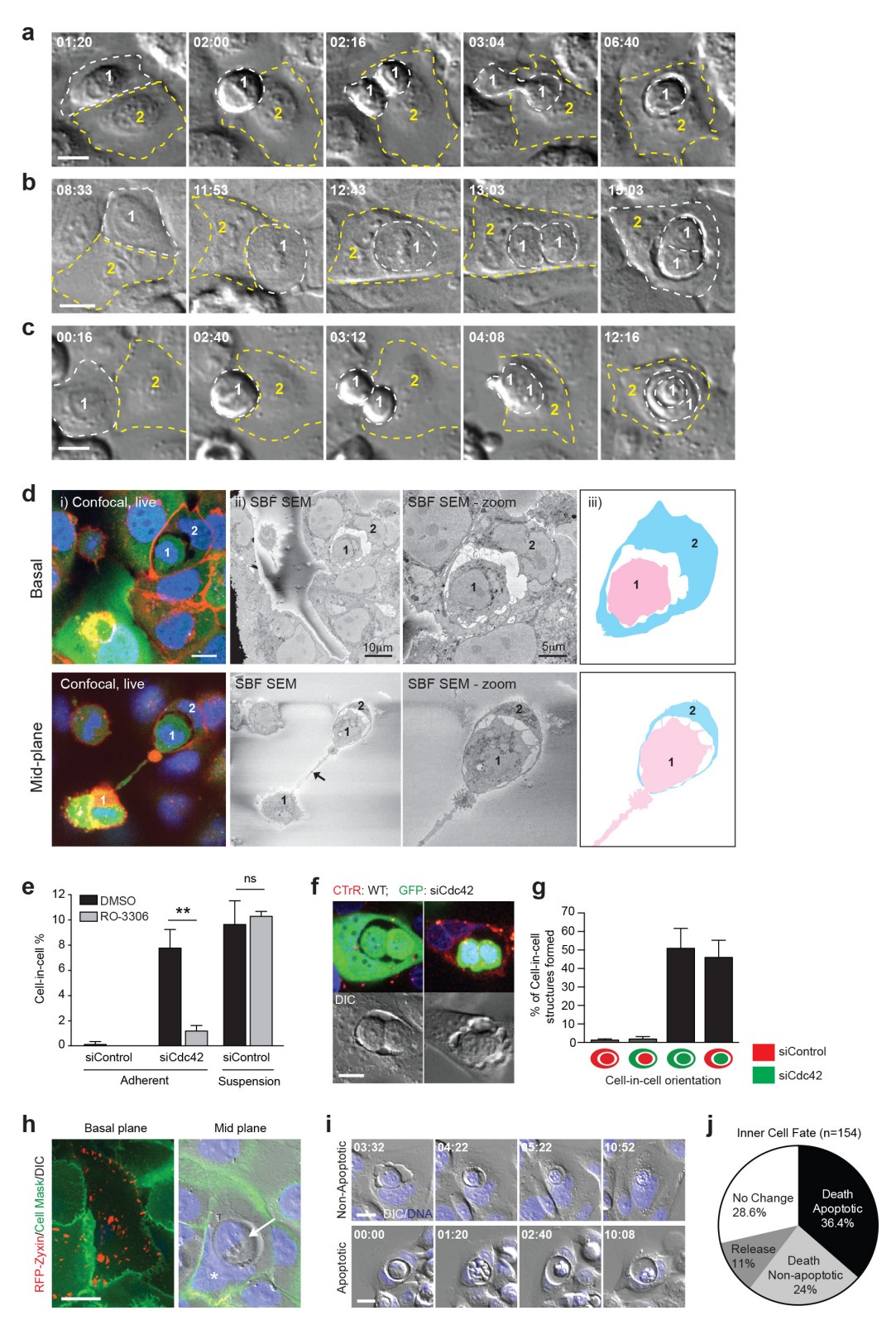

**Figure 2.** Mitosis drives entotic cell cannibalism in adherent cells. (**a–c**) Cdc42-depleted 16HBE cells were analysed by timelapse microscopy. Three different configurations of cell-in-cell formation are shown with timestamps (hr:min). In each case, a mitotic cell (Cell 1, outlined white) enters an adherent entotic host (Cell 2, outlined yellow). Scale bars = 10 µm. (**d**) Cdc42-depleted 16HBE cells were analysed by 3D-CLEM. (i) Live confocal sections from basal and mid-planes of a forming cell-in-cell structure, stained for plasma membrane (red), cell body (green) and DNA (blue). A mitotic

*Figure 2 continued on next page*

*Figure 2 continued*

daughter (Cell 1) is shown internalising into an adherent neighbour (Cell 2). Scale bar = 10 μm. (ii) Corresponding serial blockface scanning electron microscopy (SBF-SEM) images of the same forming structure. The arrowhead marks the midbody between daughter cells. (iii) Cartoon outline of cell-in-cell structure from (ii). (e) Quantification of cell-in-cell formation among control and Cdc42-depleted 16HBE cells, under adherent or suspension conditions, in the presence or absence of RO-3306 (5 μM), a Cdk1 inhibitor that induces G2/M arrest. >200 cells were counted per sample/experiment, across three separate experiments. Error bars denote mean±SEM. **p<0.002; ns = no significant difference, t-test. (f) Representative confocal and DIC images of adherent cell-in-cell structures in wild-type-16HBE (red) and Cdc42-depleted GFP-16HBE (green) co-cultures. Scale bar = 15 μm. (g) Quantification of cell-in-cell formation between WT and Cdc42-depleted 16HBE co-cultures as described in (f). >50 cell-in-cell structures were imaged per condition/experiment, across three separate experiments. Error bars denote mean±SEM. (h) Cdc42-depleted 16HBE cells were mixed with wild-type cells expressing RFP-zyxin, incubated for 3 days then stained for plasma membrane (green) and DNA (blue) and imaged by live confocal and DIC microscopy. A representative adherent cell-in-cell structure is shown, with basal and mid-plane sections presented; asterix = host cell nucleus, arrowhead = internalised cell. Scale bar = 15 μm. (i) Inner cell fate was analysed in Cdc42-depleted adherent 16HBE entotic structures, stained for DNA (blue). Representative timelapse series show non-apoptotic and apoptotic inner cell death; timestamps = (hr:min). Scale bar = 10 μm. (j) Quantification of inner cell fate over 20 hr in adherent Cdc42-depleted 16HBE structures. 154 cell-in-cell structures were analysed over three independent experiments.

The following source data is available for figure 2:

**Source data 1.**

different phases of mitosis, in live cells stained for plasma membrane and DNA (*Figure 3b*). Consistent with previous reports, we observed that Cdc42-depletion can misorient the plane of division (*Jaffe et al., 2008*; *Roignot et al., 2013*), while control cells divide parallel to the substrate, the plane of division in Cdc42-depleted cells is randomised. However, this phenotype seems unlikely to drive cell-in-cell formation, as aPKC-depletion promotes similar spindle misorientation (*Durgan et al., 2011*), with no detectable effect on entosis under these conditions (*Figure 3—figure supplement 1*). Interestingly, we also identified additional, unanticipated changes in mitotic morphology upon Cdc42-depletion (*Figure 3b–d*). Like other polarised epithelia (*Reinsch and Karsenti, 1994*), WT-16HBE cells bulge as they enter prometaphase, but retain both cell-cell and cell-matrix contacts throughout mitosis (*Figure 3b*, upper panel). In contrast, from prometaphase onwards, Cdc42-depleted cells exhibit reduced spreading, associated with a diminished adhesive surface area, and dramatic rounding, a closely related phenomenon (*Marchesi et al., 2014*) (*Figure 3b*, lower panel). These morphological changes can be quantified by calculating height/length, as a measure of cell spreading (*Figure 3c*), and circularity, as a score of roundness (*Figure 3d*). Across multiple reagents and experiments, Cdc42-depletion consistently and significantly augments mitotic deadhesion and rounding. These data uncover a novel role for Cdc42 in controlling mitotic morphology in polarised epithelial cells.

## Cdc42 regulates cortical RhoA activity in mitotic cells

We next considered the molecular mechanisms through which Cdc42 may regulate mitotic morphology and entosis, with RhoA emerging as a compelling candidate player. RhoA, another of the major small GTPases, has been implicated in mitosis (*Chircop, 2014*), mitotic rounding (*Maddox and Burridge, 2003*; *Matthews et al., 2012*) and cell-in-cell formation

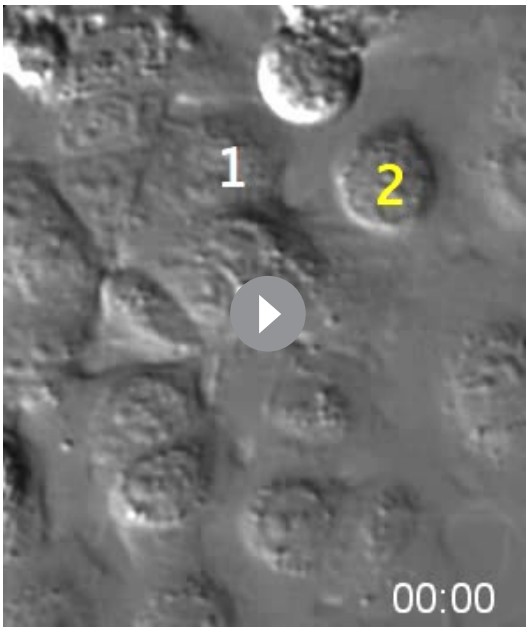

**Video 1.** Mitosis-driven entosis in adherent Cdc42-depleted 16HBE cells. DIC images from Widefield timelapse. Cell 1 engulfed by cell 2 post mitosis.

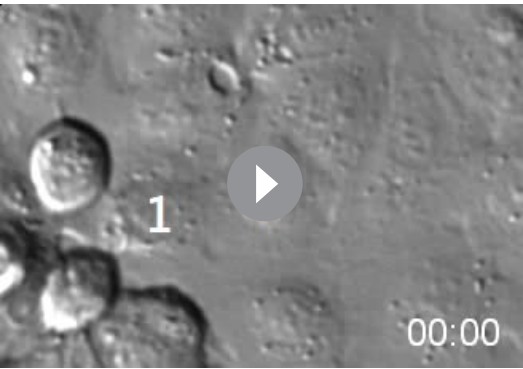

**Video 3.** Mitosis-driven entosis in adherent Cdc42-depleted 16HBE cells. DIC images from Widefield timelapse. Cell 1 daughters engulf each other and are then engulfed by cell 2.

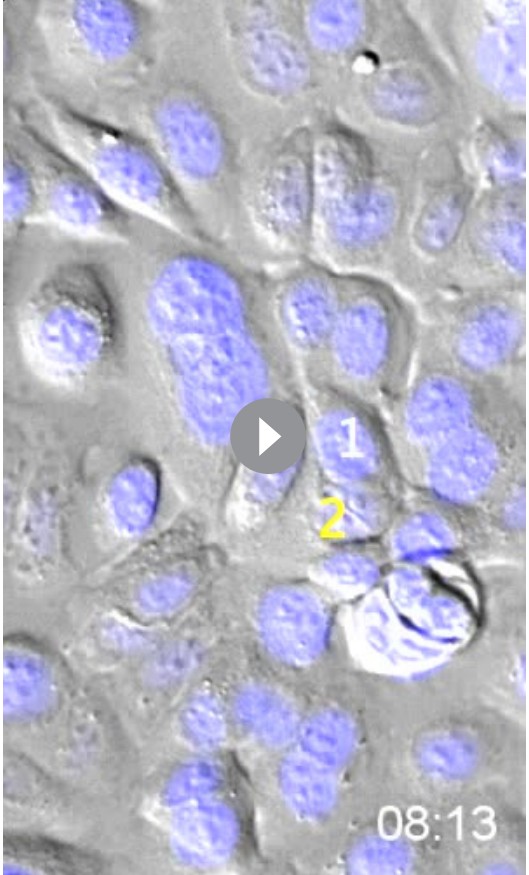

**Video 2.** Mitosis-driven entosis in adherent Cdc42-depleted 16HBE cells. DIC images from Widefield timelapse. Cell 1 is engulfed by cell 2 during mitosis.

(*Overholtzer et al., 2007*) through previous studies. To investigate a possible role here, the spatiotemporal activation of RhoA was assessed using a FRET-based biosensor, RhoA-FLARE (*Sun et al., 2014b*; *Pertz et al., 2006*). Control or Cdc42-depleted 16HBE cells expressing the RhoA biosensor were subjected to live confocal imaging for CFP (FRET donor) and YFP (FRET acceptor), as well as DNA (Hoechst) and DIC (*Figure 4a*); RhoA activation is proportional to the FRET/CFP emission ratio. While control 16HBE cells show a relatively low level of RhoA activity during metaphase, a clear enrichment of active RhoA is frequently observed at the metaphase cortex among Cdc42-depleted cells (37%; *Figure 4b*). Related to this, an increase in cortical actin can also be observed among Cdc42-depleted, metaphase cells (*Figure 4c*). Together, these data indicate that Cdc42 constrains mitotic RhoA activation in polarised epithelial cells, and accordingly, that loss of Cdc42 permits overactivation of cortical RhoA during metaphase.

## RhoA activity controls mitotic morphology

To test whether deregulated RhoA activation influences mitotic morphology, live Cdc42-depleted cells were treated with a cell permeable form of C3 Transferase, a toxin that selectively ribosylates and inactivates RhoA/B/C (*Barbieri et al., 2002*), to block downstream signalling (*Figure 4d*). Although mitotic cells do remain somewhat rounded in the presence of this toxin, inhibition of the Rho proteins has a clear impact on mitotic spreading, yielding a significant increase in basal area, and thus partially reversing the effect of Cdc42-depletion (*Figure 4e*). These data are consistent with previous studies that have implicated RhoA activity in mitotic cell retraction, rigidity and rounding (*Maddox and Burridge, 2003*; *Matthews et al., 2012*), along with its targets ROCK (*Maddox and Burridge, 2003*; *Meyer et al., 2011*) and myosin (*Matthews et al., 2012*). To explore this observation further, live Cdc42-depleted cells were treated with additional pathway inhibitors, and suppression of ROCK (Y-27632) or myosin (blebbistatin) activity similarly rescued mitotic cell spreading in Cdc42-depleted cells (*Figure 4d–e*). Together, these data indicate that aberrant activation of a Rho/ROCK/myosin cascade during mitosis can drive enhanced retraction and rounding. Based on these findings, we would predict that RhoA inhibition may also suppress mitotic entosis upon Cdc42-depletion, by reverting these induced changes in mitotic morphology. Unfortunately, it is not possible to demonstrate this link unambiguously, because RhoA inhibition will block entotic

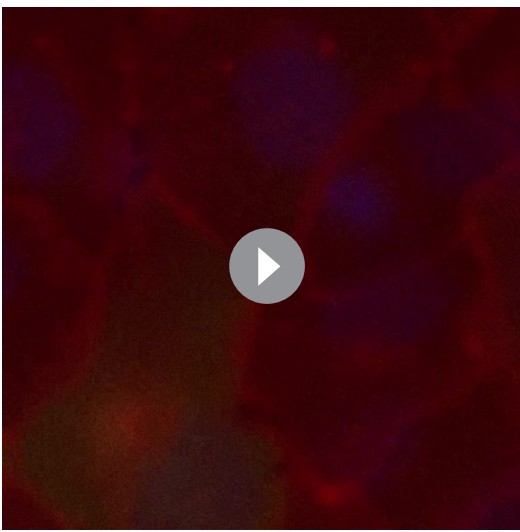

**Video 4.** Live cell confocal z-stack of a forming cell-in-cell structure in Cdc42-depleted adherent 16HBE cells. Cells are stained with CellTracker green (cell body), CellMASK, red (membrane) and Hoechst (DNA, blue). DOI: 10.7554/eLife.27134.010

cell-in-cell formation regardless of its trigger, due to downstream effects on myosin contractility (*Overholtzer et al., 2007*) and actin dynamics (*Purvanov et al., 2014*). As such, while we can clearly implicate RhoA in both mitotic rounding and cell-in-cell formation, we cannot definitively dissect the process of mitotic entosis further by targeting RhoA alone.

## Enhanced mitotic deadhesion and rounding can induce entosis: Rap1

As an alternative approach to determine whether mitotic deadhesion and rounding is sufficient to drive entosis, mitotic morphology was manipulated through other means. Rap1 is yet another small GTPase, which, importantly here, is known to control post-mitotic spreading (*Dao et al., 2009*; *Lancaster et al., 2013*). Consistent with this, we confirm that expression of DN-Rap1 reduces spreading and increases rounding during 16HBE division (*Figure 5a–d*). Strikingly, inhibition of Rap1, like Cdc42, also induces adherent entosis, albeit at a lower level. Cell-in-cell formation among Rap1-inhibited cells occurs during or shortly after mitosis (*Figure 5e–g*, *Video 6*) and is inhibited by G2 arrest (Cdk1i). These data are consistent with a model in which enhanced mitotic deadhesion and rounding drive subsequent cell-in-cell formation. In light of these data, we next asked whether mitosis might promote entosis by simply providing an alternative route to matrix deadhesion in one cell of the pair (the dividing, internalised cell). To address this, we tested whether detached, interphase cells can similarly penetrate matrix-attached hosts. WT-16HBE cells were labelled green, detached and then either overlaid onto wild-type monolayers (adherent hosts) or maintained in suspension for 8 hr; the resulting cell-in-cell structures were scored (*Figure 5h–i*). Strikingly, while detached interphase cells undergo efficient entosis in suspension (~8%), penetration of adherent hosts is barely detectable. These data indicate that simple loss of matrix-attachment is insufficient to drive penetration of an adherent host. Consistent with this, entosis has not been observed among 16HBE-monolayers upon depletion of matrix-adhesion proteins (e.g. $\beta$1-integrin; *Figure 5—figure supplement 1*). Together, these data indicate that mitotic deadhesion and rounding, which can be augmented by inhibition of Cdc42 or Rap1, can drive the entotic penetration of an adherent host cell, while matrix-detachment during interphase cannot. These findings infer important mechanistic differences between cell cannibalism under detached and adherent conditions. We conclude that the distinctive biophysical changes associated with mitosis are an essential requirement when entosis occurs in an adherent host.

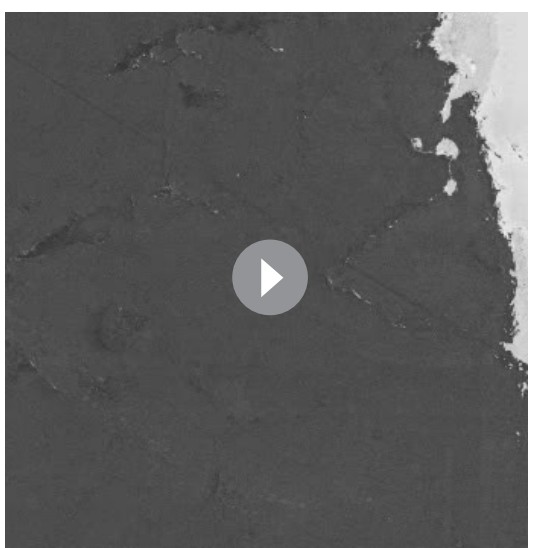

**Video 5.** Corresponding Serial Block Face SEM z-stack of forming cell-in-cell structure in *Video 4*. DOI: 10.7554/eLife.27134.011

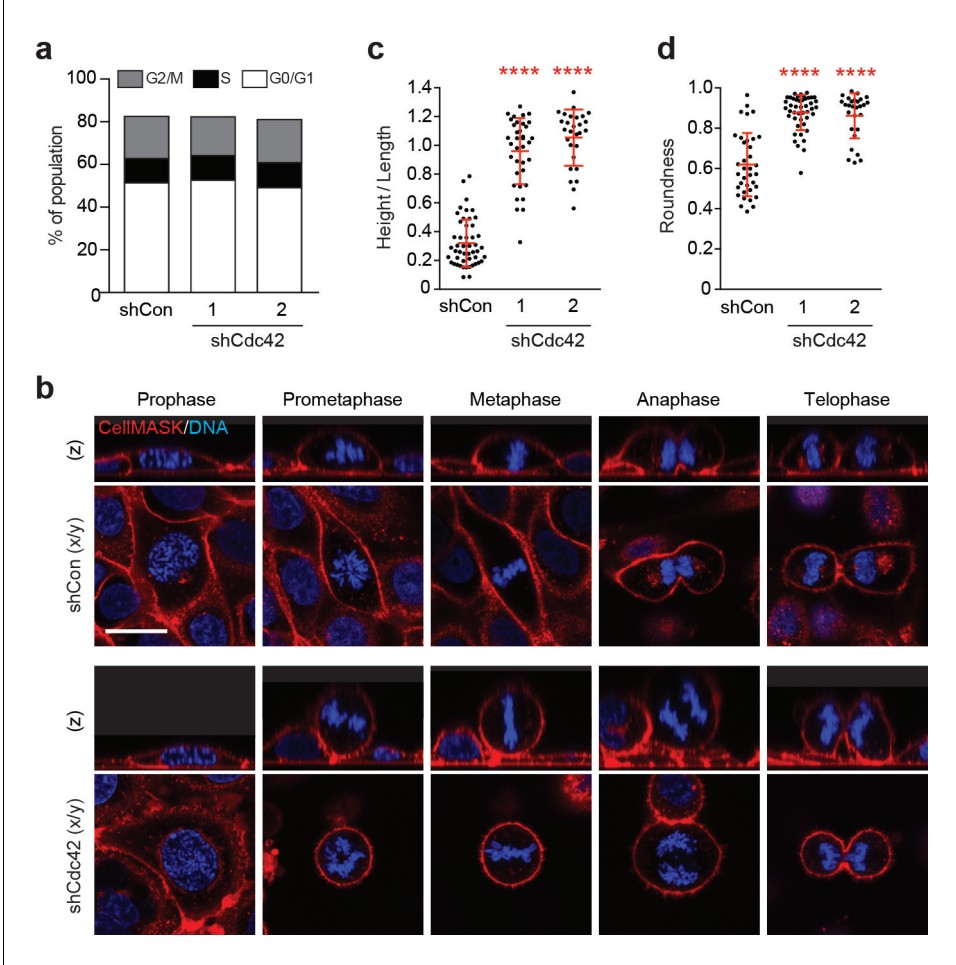

**Figure 3.** Cdc42 controls mitotic deadhesion and rounding in polarised epithelial cells. (**a**) Control and Cdc42-depleted 16HBE cells were fixed and stained with propidium iodide. DNA content was analysed by FACS. (**b**) Control and Cdc42-depleted 16HBE cells were stained for plasma membrane (red) and DNA (blue), and analysed by live confocal microscopy. Representative sections and z-stacks of different phases of mitosis are shown. Scale bar = 20 μm. Quantification of (**c**) mitotic spreading (cell height/length) and (**d**) mitotic rounding (where 1 = a perfect circle) in control and Cdc42-depleted cells. >10 metaphase cells were imaged per sample/experiment, across three independent experiments. Error bars denote mean±SD. ****p<0.0001, Mann-Whitney U test.

The following source data and figure supplement are available for figure 3:

**Source data 1.**

**Figure supplement 1.** Cdc42 controls adherent cell-in-cell formation, but aPKC does not.

## Mitosis-induced entosis occurs constitutively in adherent cancer cell lines and human tumours with pleiotropic effects

Given that tumour cells are prone to undergo deregulated division and that mitotic rounding is proposed to be of relevance in cancer (*Cadart et al., 2014*), we hypothesised that transformed cells may undergo mitotic entosis constitutively. To test this, adherent cancer cell lines were examined by fixed and live microscopy. Interestingly, MCF7 (breast) and HCT116 (colorectal) cells were found to undergo adherent entosis under basal conditions. Importantly, cell-in-cell formation is coincident with cell division, as shown by timelapse microscopy (*Figure 6a–b*, *Videos 7–8*), and is largely dependent on progression through mitosis, in both 2D and 3D culture (*Figure 6c–d*). These data indicate that mitosis drives constitutive adherent entosis in certain transformed tumour cell lines and

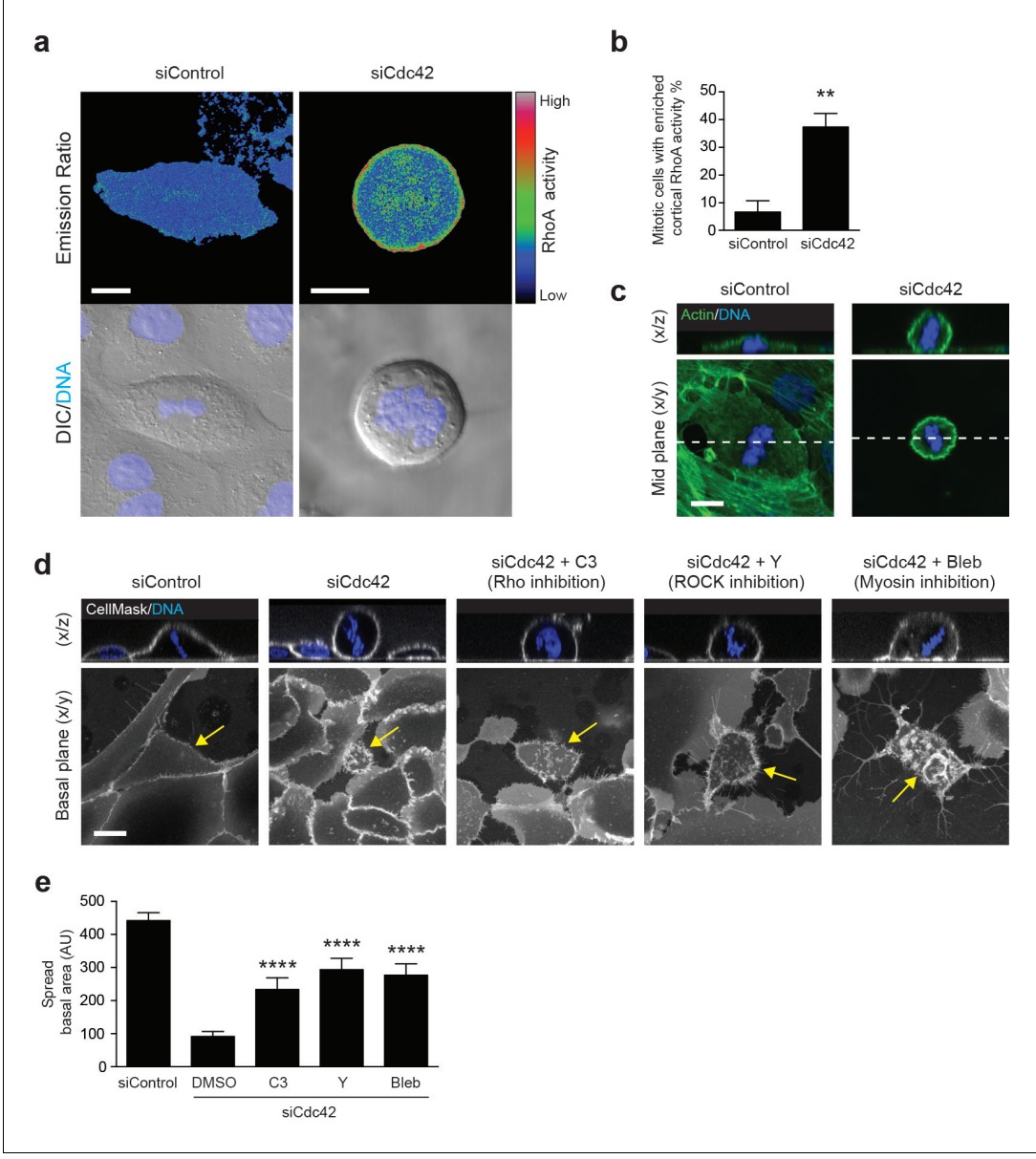

**Figure 4.** RhoA activity is spatiotemporally regulated by Cdc42 and controls mitotic spreading. (a) 16HBE cells expressing a RhoA FRET biosensor were treated with siControl or siCdc42. Three days later, cells were subjected to live confocal imaging for CFP (FRET donor), YFP (FRET acceptor), DNA and DIC. RhoA activity is represented by the FRET/CFP emission ratio. Scale bars = 10 μm. (b) RhoA activity was measured in >30 metaphase cells per condition, across four independent experiments, and cortical enrichment of active RhoA was scored. **p<0.003, t-test. (c) Control or Cdc42-depleted 16HBE cells were fixed and stained for actin (green) or DNA (blue), and metaphase cells were imaged by IF/confocal. Representative sections and z-stacks are shown. Scale bar = 10 μm. (d) 16HBE cells were treated with siControl or siCdc42 and incubated for 3 days. Cdc42-depleted cells were then incubated with C3 (Rho inhibitor; 1 μg/ml), Y-27632 (ROCK inhibitor; 10 μM) or Blebbistatin (myosin inhibitor; 100 μM) for a further 4 hr. Live cells were stained for plasma membrane (white) and DNA (blue) and imaged by IF/confocal to assess metaphase morphology; scale bar = 15 μm. Representative z-stacks and basal sections are shown. (e) The spread basal area of each metaphase cell was measured. >15 cells were scored/condition, across three independent experiments. ****p<0.0001.

The following source data is available for figure 4:

**Source data 1.**

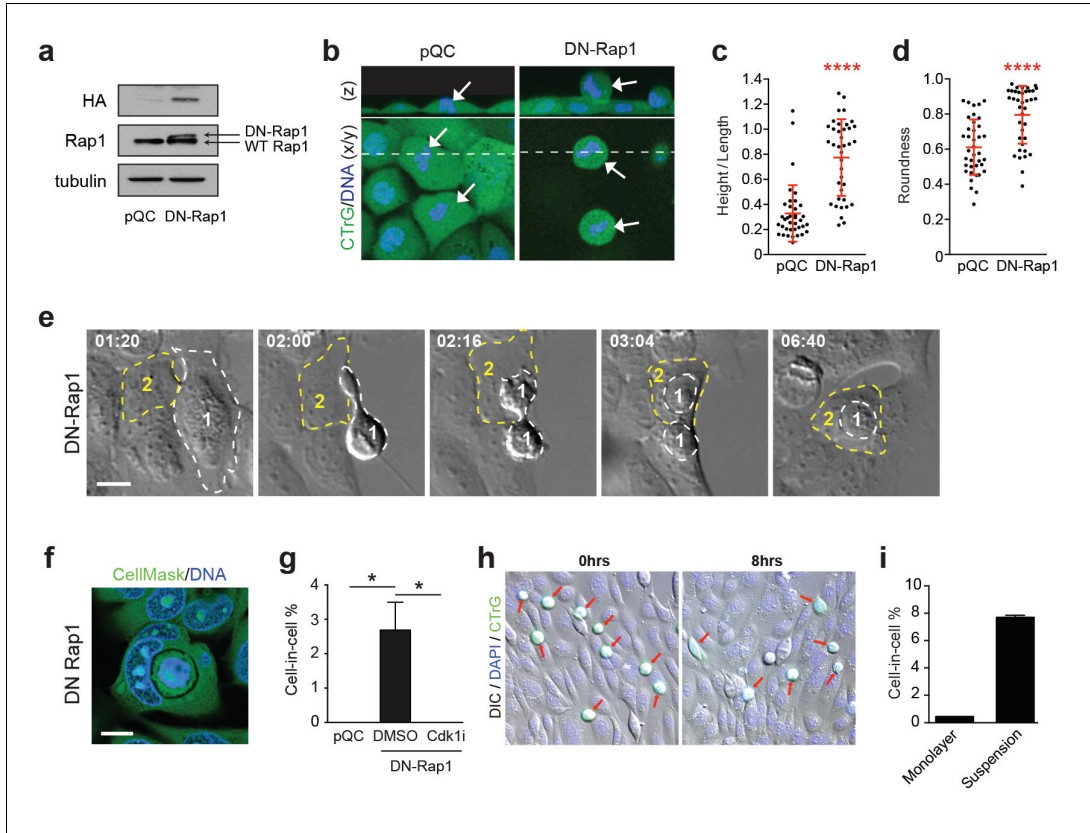

**Figure 5.** Enhanced mitotic deadhesion and rounding can drive entosis. (a) Control (pQC) and DN-Rap1-HA expressing 16HBE cells were probed for HA, Rap1 and tubulin by western blot. (b) Control and DN-Rap1 expressing 16HBE cells were stained for cell body (green) and DNA (blue) and analysed by live confocal microscopy. Representative midplane x/y, and z sections through the dashed line, are presented. Arrowheads indicate metaphase cells, as identified by nuclear morphology. Quantification of (c) mitotic spreading (cell height/length) and (d) mitotic rounding (where 1 = a perfect circle) in control and DN-Rap1 16HBE cells. >10 metaphase cells were imaged per sample/experiment, across three independent experiments. Error bars denote mean±SD. ****p<0.0001, Mann-Whitney U test. (e) DN-Rap1 expressing 16HBE cells were analysed by timelapse microscopy. A mitotic cell (Cell 1) is outlined in white, the adherent entotic host (Cell 2) in yellow. Timestamps are indicated (hr: min) and scale bar = 10 µm. (f) Representative confocal image of an adherent cell-in-cell structure in DN-Rap1 expressing 16HBE cells, fixed and stained for the cell body (green) and DNA (blue). Scale bar = 10 µm. (g) Quantification of cell-in-cell formation in adherent control and DN-Rap1 cells, treated −/+ a Cdk1 inhibitor that induces G2/M arrest (5 µM RO-3306; Cdk1i). >250 cells were counted per sample/experiment, across three separate experiments. Error bars denote mean±SEM. *p<0.03, t-test. (h) Representative images from a co-culture of suspension wild type 16HBE cells, labelled with CellTracker (green), overlaid on an existing monolayer of wild type 16HBE cells stained for DNA (blue). Co-cultures were monitored for 8 hr by timelapse microscopy. DIC/IF images are shown at time 0 and 8 hr; the detached population (green) are highlighted with red arrows; these cells can persist throughout the timecourse and very rarely penetrate adherent hosts. (i) Quantification of cell-in-cell formation under adherent conditions as described in (h), and under suspension conditions. >300 cells were counted per sample/experiment, across three independent experiments. Error bars denote mean±SEM.

The following source data and figure supplement are available for figure 5:

**Source data 1.**

**Figure supplement 1.** Loss of β1 integrin (ITGB1) does not induce adherent cell-in-cell formation.

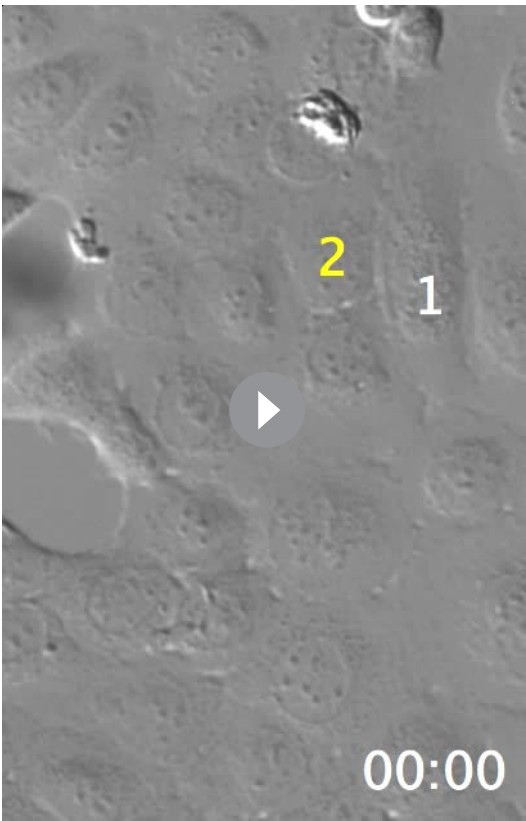

**Video 6.** Mitosis-driven entosis in adherent 16HBE cells expressing DN-Rap1. DIC images from Widefield timelapse. A daughter of cell 1 is engulfed by cell 2. DOI: 10.7554/eLife.27134.020

raise the interesting possibility that cell division may promote cell cannibalism in the proliferative environment of a tumour. To investigate this more directly, a tissue microarray of 75 human breast invasive ductal carcinomas was analysed (*Figure 6e–g*, *Supplementary file 1*). Each tumour core was stained for DNA, β-catenin and p-Histone-H3 (pS10; a mitotic marker), and then imaged in full and scored for mitotic activity (mitotic cells/core) and entosis (cell-in-cell structures/core). 31/75 ductal carcinomas were positive for entosis, and among these, mitotic activity positively correlates with cell-in-cell formation, with statistical significance (*Figure 6g*). These data are consistent with the notion that mitosis may drive entotic cell-in-cell formation in vivo. During conventional entosis, the inner cell is typically killed (*Overholtzer et al., 2007*; *Florey et al., 2011*; *Wan et al., 2012*), while the outer cell is rendered prone to division failure (*Krajcovic et al., 2011*; *Wan et al., 2012*). To determine whether similar anti- and pro-tumorigenic consequences accompany mitotic entosis among adherent cancer cells, the outcomes were followed by timelapse and IF/confocal microscopy. Similar to suspension conditions, cell death is the predominant fate for internalised cells following mitotic entosis, which can occur by either non-apoptotic and apoptotic means (*Figure 6h–i*). Host cell division failure was also analysed by scoring for multinucleation. Again, similar to suspension entosis, host cells are more frequently multi-nucleated than surrounding single cells, suggesting that abscission can be disrupted by entosis in adherent, as well as matrix-detached, conditions (*Figure 6j–k*). Together, these data establish that mitosis-induced entosis occurs basally in adherent cancer cell lines and human breast carcinomas. Mitosis-induced entosis can drive inner cell killing, a potentially tumour suppressive effect, but also promotes host cell multi-nucleation, a route to tumour-promoting genomic instability. These findings uncover an intriguing relationship between cell division and cannibalism, of potential functional significance during tumour development and evolution.

## Taxane treatment promotes mitosis-induced entosis

Many cancers are treated chemotherapeutically with taxanes (e.g. Paclitaxel), which can induce prometaphase arrest, multipolar division and cell death (*Zasadil et al., 2014*). As such, we hypothesised that taxane treatment might influence cell cannibalism by modulating mitotic entosis. To test this, 16HBE cells were incubated in the presence or absence of taxol, then imaged by timelapse or IF/confocal microscopy (*Figure 7a–c*, *Video 9*). As expected, taxol treatment increases mitotic index under these conditions, arresting the cells in prometaphase (in contrast to Cdk1i which arrests at G2/M, inhibiting entry into mitosis). Importantly, taxol treatment also consistently increases cell-in-cell formation under these conditions, in line with an induction of mitotic entosis. This effect can be phenocopied with additional drugs that inhibit mitotic progression through different mechanisms, including nocodazole (microtubule assembly inhibitor) and STLC (mitotic kinesin inhibitor), suggesting that entosis occurs as a consequence of prometaphase arrest rather than microtubule stabilisation. Strikingly, we also find that taxol, nocodazole and STLC significantly enhance mitotic deadhesion and rounding (*Figure 7d–f*), thus independently supporting the model that changes in mitotic morphology are closely associated with adherent cell-in-cell formation. To investigate

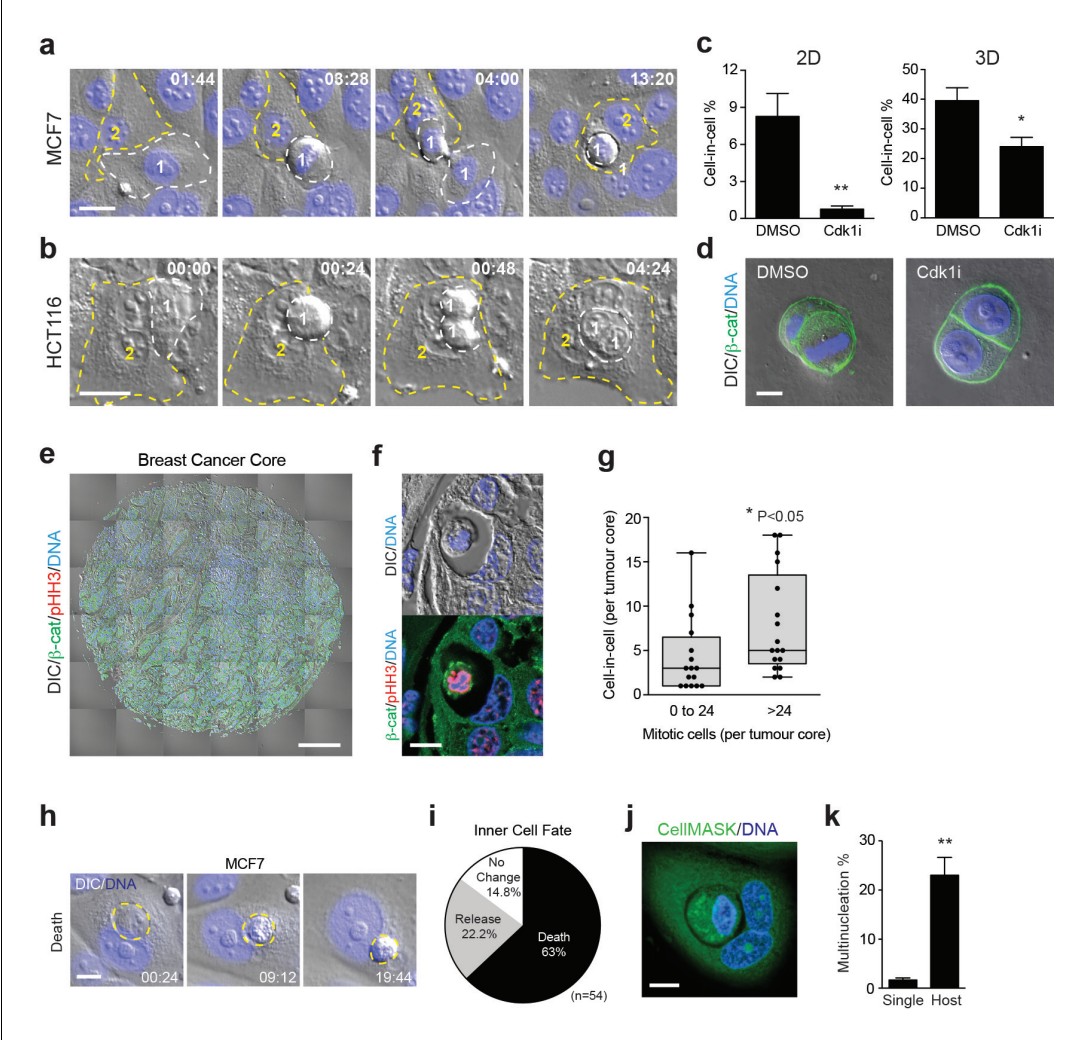

**Figure 6.** Mitosis-induced entosis occurs in cancer cell lines and human tumours, with pleiotropic effects. (**a–b**) Adherent cancer cell lines MCF7 (breast) and HCT116 (colorectal) were analysed by timelapse microscopy; representative cell-in-cell formation events are shown. In each case, a mitotic cell (Cell 1, outlined in white) is internalised by an adherent neighbour (Cell 2, outlined in yellow). Timestamps are shown (hr:min) and scale bar = 20 µm. (**c**) Quantification of cell-in-cell formation in adherent MCF7 cells cultured in 2D or 3D, treated +/− a Cdk1 inhibitor that induces G2/M arrest (5 µM RO-3306; Cdk1i). >300 cells for 2D and >50 cells for 3D were counted per sample/experiment, across three separate experiments. Error bars denote mean±SEM. **p<0.002; *p<0.02, t-test. (**d**) MCF7 cells seeded in 3D matrigel were treated −/+5 µM RO-3306 overnight, then fixed and stained for β-catenin (green) and DNA (blue). Two to four cell cysts were imaged to assess mitotic status and cell-in-cell formation. Representative sections are shown; scale bar = 10 µm. (**e**) Human breast invasive ductal carcinoma. A tumour microarray was stained for β-catenin (green), pS10-Histone H3 (red, mitotic marker) and DNA (blue) and imaged in full by DIC and IF/confocal. A tiled confocal image is presented for one core. Scale bar = 200 µm. (**f**) Entosis in a human invasive breast ductal carcinoma. A representative cell-in-cell structure is shown by DIC and IF/confocal; notably the inner cell is mitotic as judged by pHH3. Scale bar = 10 µm. (**g**) Quantification of mitotic index and cell-in-cell formation among human breast invasive ductal carcinomas. The median number of mitotic cells/core is 24. *p<0.05, Mann-Whitney test. (**h**) Representative timelapse series showing inner cell death in an MCF7 cell-in-cell structure, stained for DNA (blue). The internalised cell is outlined in yellow; its corpse shrinks over time. Timestamps are indicated (hr:min) and scale bar = 10 µm. (**i**) Quantification of inner cell fate in MCF7 entotic structures over 20 hr. Fifty-four cell-in-cell structures were analysed over three independent experiments. (**j**) Representative image of a multinucleated, entotic host cell in adherent MCF7 stained for cell body (green) and DNA (blue). Scale bar = 10 µm. (**k**) Quantification of MCF7 multinucleation in single cells versus entotic hosts cells. Error bars denote mean±SEM across three independent experiments. **p<0.008, t-test.

The following source data is available for figure 6:

**Source data 1.**

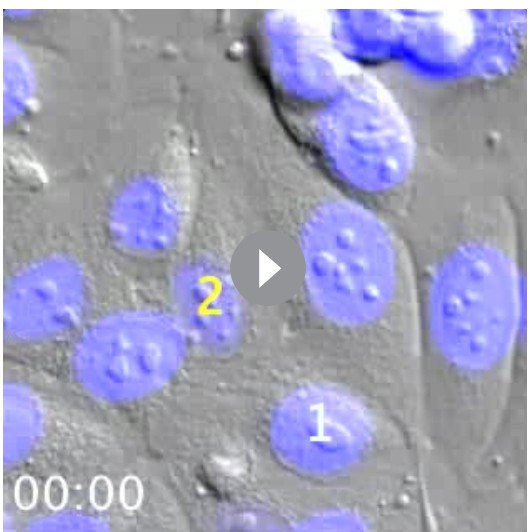

**Video 7.** Mitosis-driven entosis in adherent MCF7 cells. DIC and DNA(Hoechst) images from Widefield timelapse. A daughter of cell 1 is engulfed by cell 2.

whether taxane-induced cell cannibalism may be of chemotherapeutic significance, MCF7 breast cancer cells were examined. In both cultured cells (*Figure 7g–h*) and mouse xenograft models (*Figure 7i–l*), 24 hr taxol treatment promotes a significant increase in mitotic index, and a corresponding induction of cell-in-cell formation, consistent with mitosis-induced entosis. Notably, as taxol is reported to inhibit MCF7 entosis in suspension (*Xia et al., 2014*), these data further suggest that the mechanisms of detached versus adherent cell-in-cell formation are quite distinct. Together, our findings establish that taxane treatment enhances mitotic deadhesion and rounding and promotes cell cannibalism through entosis, and reinforce the conclusion that changes in mitotic morphology can drive adherent entosis. Moreover, these findings uncover some intriguing and unconventional new effects of taxane treatment, of potential chemotherapeutic interest.

## Discussion

Cell cannibalism has been observed within tumours for over a century (*Overholtzer and Brugge, 2008*; *He et al., 2013*; *Lozupone and Fais, 2015*), but its underlying mechanisms and functional consequences remain to be fully understood. Entosis is a form of homotypic epithelial cell cannibalism that is typically triggered by matrix-deadhesion, and which proceeds through junction formation and the generation of actomyosin contractility, culminating in cell engulfment and killing (*Overholtzer et al., 2007*). This process is regulated at various stages by Rho-family small GTPases, which govern the cytoskeletal and junctional components upon which entosis depends (*Overholtzer et al., 2007*; *Sun et al., 2014a*; *Purvanov et al., 2014*).

This study set out to address whether Cdc42, a master regulator of epithelial biology (*Jaffe and Hall, 2005*; *Heasman and Ridley, 2008*; *Joberty et al., 2000*; *Martin-Belmonte et al., 2007*; *Jaffe et al., 2008*; *Wallace et al., 2010*; *Roignot et al., 2013*), controls entosis. Surprisingly, despite its well-documented control of the cytoskeleton, polarity and epithelial junctions (*Etienne-Manneville, 2004*), Cdc42 has little effect on entosis among cells cultured in suspension. Unexpectedly, however, we found that loss of Cdc42 can in fact promote robust cell-in-cell formation among adherent cells. This result was very surprising, because entosis is not expected to occur under these conditions (*Overholtzer et al., 2007*), but, this process otherwise bears its hallmarks, involving cell-cell contacts (*Sun et al., 2014b*; *Wang et al., 2015*), polarised actomyosin (*Sun et al., 2014a*), autophagy proteins (*Florey et al., 2011*) and lysosomes (*Krajcovic et al., 2013*). Notably, some similar, adherent engulfment events have been noted previously (*Lai et al., 2010*; *Abreu and Sealy, 2012*), and in the present study, we clarify that

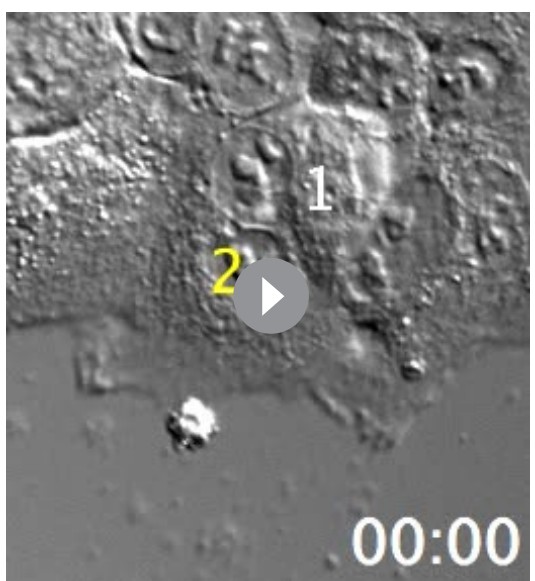

**Video 8.** Mitosis-driven entosis in adherent HCT116 cells. DIC images from Widefield timelapse. Cell 1 is engulfed by cell 2 during mitosis.

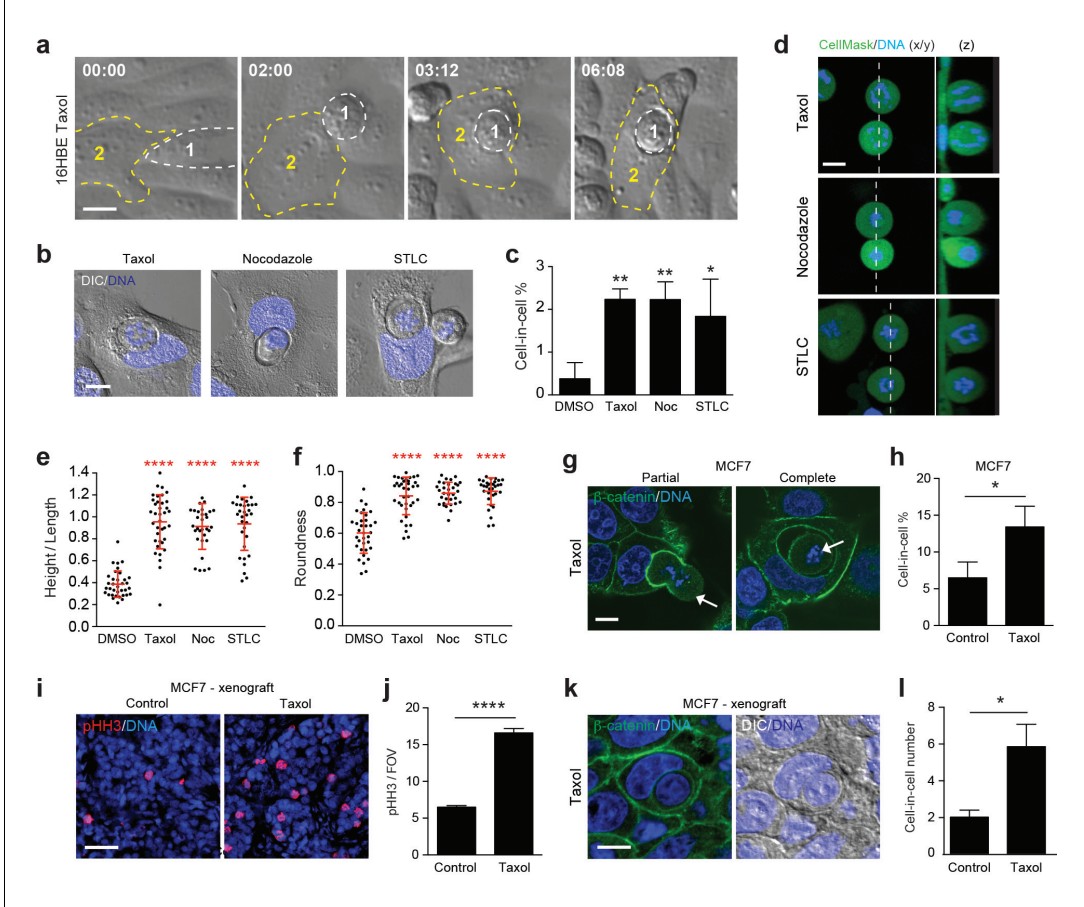

**Figure 7.** Paclitaxel/taxol treatment promotes mitotic deadhesion, rounding and entosis. (a) Representative timelapse images of adherent 16HBE cells treated with 1 μM taxol. Cell 1 (outlined white) rounds up in prometaphase and subsequently penetrates an adherent interphase neighbour (Cell 2, outlined yellow). Timestamps are shown (hr:min) and scale bar = 10 μm. (b) Representative confocal/DIC images of adherent cell-in-cell structures in 16HBE treated with taxol (1 μM), nocodazole (100 ng/ml) or STLC (20 μM) for 24 hr. Cells were stained for DNA (blue), scale bar = 10 μm. (c) Quantification of drug-induced cell-in-cell formation. >150 cells were counted per sample/experiment, across three separate experiments. Error bars denote mean±SEM. **p<0.002; *p<0.02, t-test. (d) Mitotic morphology of 16HBE cells treated with taxol (1 μM), nocodazole (100 ng/ml) or STLC (20 μM) for 24 hr. Live cells were stained for cell body (green) and DNA (blue). Midplane x/y, and z sections through the dashed line, are presented. Scale bar = 10 μm. Quantification of (e) mitotic spreading (cell height/length) and (f) mitotic rounding (where 1 = a perfect circle) in control and drug-treated cells. >15 cells were counted per sample/experiment, across three separate experiments. Error bars denote mean±SD. ****p<0.0001, Mann-Whitney U test. (g) Representative confocal images of partially and completely formed cell-in-cell structures in MCF7 cells treated with taxol (1 μM), and stained for β-catenin (green) and DNA (blue). The arrowheads point to prometaphase arrested cells internalised by adherent, interphase neighbours. Scale bar = 10 μm. (h) Quantification of taxol-induced entosis in MCF7. >150 cells were counted per sample/experiment, across three separate experiments. Error bars denote mean±SEM. *p<0.04, t- test. (i) Confocal images of MCF7 xenografts treated −/+ taxol for 24 hr and stained for phospho-Histone H3 (pHH3, red) and DNA (blue). Scale bar = 50 μm. (j) Quantification of pHH3-positive, mitotic cells in MCF7 mouse xenografts treated −/+ taxol for 24 hr. ****p<0.0001, t-test. (k) Representative confocal and DIC images of an entotic cell-in-cell structure in a taxol-treated MCF7 xenograft, stained for β-catenin (green) and DNA (blue). Scale bar = 10 μm. (l) Quantification of cell-in-cell formation in MCF7 mouse xenografts treated −/+ taxol for 24 hr. *p<0.01, t-test.

The following source data is available for figure 7:

**Source data 1.**

entosis can indeed occur among matrix-attached cells, and define the associated mechanism.

Cdc42 is known to regulate cell-matrix contacts through β1-integrin in certain cancer cells (*Reymond et al., 2012*), so it is plausible to hypothesise that its depletion may weaken focal adhesions, or impair integrin signalling, to mimic detachment and so induce entosis. However, our data are not consistent with this model. Firstly, control monolayers can efficiently internalise Cdc42-

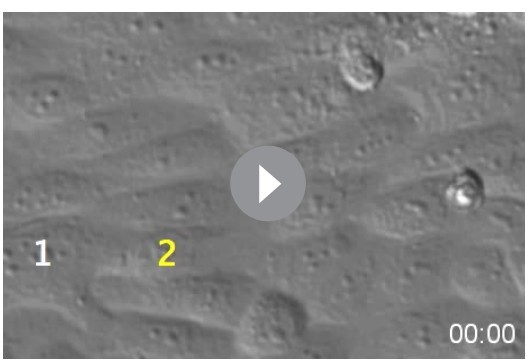

**Video 9.** Mitosis-driven entosis in adherent 16HBE cells treated with taxol (1 μM). DIC images from Widefield timelapse. Cell 2 enters mitosis and is engulfed by cell 1.

depleted neighbours, demonstrating that entotic hosts can be wild-type and therefore fully adherent. Secondly, neither enzymatic deadhesion, nor depletion of $\beta$1-integrin, is sufficient to drive one population of cells to penetrate surrounding adherent neighbours. Together, these data indicate that general changes in matrix-binding are unlikely to fully account for adherent entosis.

In this study, we identify mitosis as a novel trigger for entosis. We show that mitosis is indispensable for adherent cell-in-cell formation, with entosis occurring during, or shortly after, cell division, and requiring transit through G2/M. In contrast, mitosis is not necessary for entosis in suspension, indicating that quite distinct mechanisms operate under different growth conditions. Our findings identify an important new route to entosis and uncover an intriguing relationship between epithelial cell division and cannibalism, of particular interest in the context of a proliferative and nutrient-deprived tumour.

Cdc42 activity is known to be cell-cycle regulated (*Yoshizaki et al., 2003*) and to control certain mitotic processes, including kinetochore attachment and chromosome segregation (*Chircop, 2014*; *Yasuda et al., 2004*); however, these roles appear unrelated to cell cannibalism. We have also found little evidence to support a role for aPKC as a Cdc42 effector during adherent entosis, likely excluding mechanisms involving the polarity complex (*Joberty et al., 2000*), such as junctional remodelling (*Wallace et al., 2010*), spindle orientation (*Durgan et al., 2011*), or control of the metaphase cortex (*Rosa et al., 2015*). Instead, we have discovered an additional role for Cdc42 in regulating mitotic morphology, consistent with previous observations in NRK (*Zhu et al., 2011*) and HeLa (*Mitsushima et al., 2009*) cells. Upon Cdc42 depletion, mitotic deadhesion and rounding are significantly enhanced. These phenotypes are associated with a prominent increase in cortical RhoA activity, and can be reverted by inhibition of RhoA, or its downstream effectors ROCK and myosin, consistent with previous work (*Maddox and Burridge, 2003*; *Matthews et al., 2012*). We conclude that Cdc42 plays a novel role in the regulation of mitotic morphology in polarised epithelial cells, in a RhoA/ROCK/myosin-dependent manner. It is interesting to note that this Rho-dependent signalling axis is also required for entosis among suspension cells (*Overholtzer et al., 2007*; *Sun et al., 2014a*; *Wan et al., 2012*; *Li et al., 2015*), suggesting that some interesting mechanistic parallels exist between these alternative routes to cell cannibalism.

The augmentation of mitotic deadhesion and rounding induced by Cdc42 suppression can be phenocopied by inhibition of Rap1, or through drug-induced prometaphase arrest, and, importantly, in each case is followed by subsequent cell-in-cell formation (*Figure 8*). Putting together the data from these diverse conditions, we conclude that the distinctive biophysical changes associated with mitotic deadhesion and rounding may uniquely enable a dividing cell to penetrate an adherent epithelium, by which it is ultimately cannibalised. These findings add to the emerging concept that mitotic shape changes bear important functional consequences (*Lancaster et al., 2013*; *Cadart et al., 2014*; *Théry and Bornens, 2006*; *Gibson et al., 2011*; *Luxenburg et al., 2011*; *Kondo and Hayashi, 2013*; *Sorce et al., 2015*), of particular significance within tumours (*Sorce et al., 2015*).

We report that the process of mitosis-induced entosis is observed basally among certain cancer cell lines, establishing a broader biological incidence of this process. Moreover, we find that mitotic index positively and significantly correlates with cell-in-cell formation in human breast invasive ductal carcinomas, consistent with a pathophysiological occurrence in vivo. Importantly, as mitotic index is one of the key criteria used to stage breast cancers, our findings can in part explain the increased frequency of cell-in-cell structures among higher grade, more proliferative tumours (*Krajcovic et al., 2011*; *Gupta and Dey, 2003*; *Abodief et al., 2006*). Our study thus contributes new insights into the field of cell cannibalism in cancer, building on the emerging notion that cell-in-cell formation

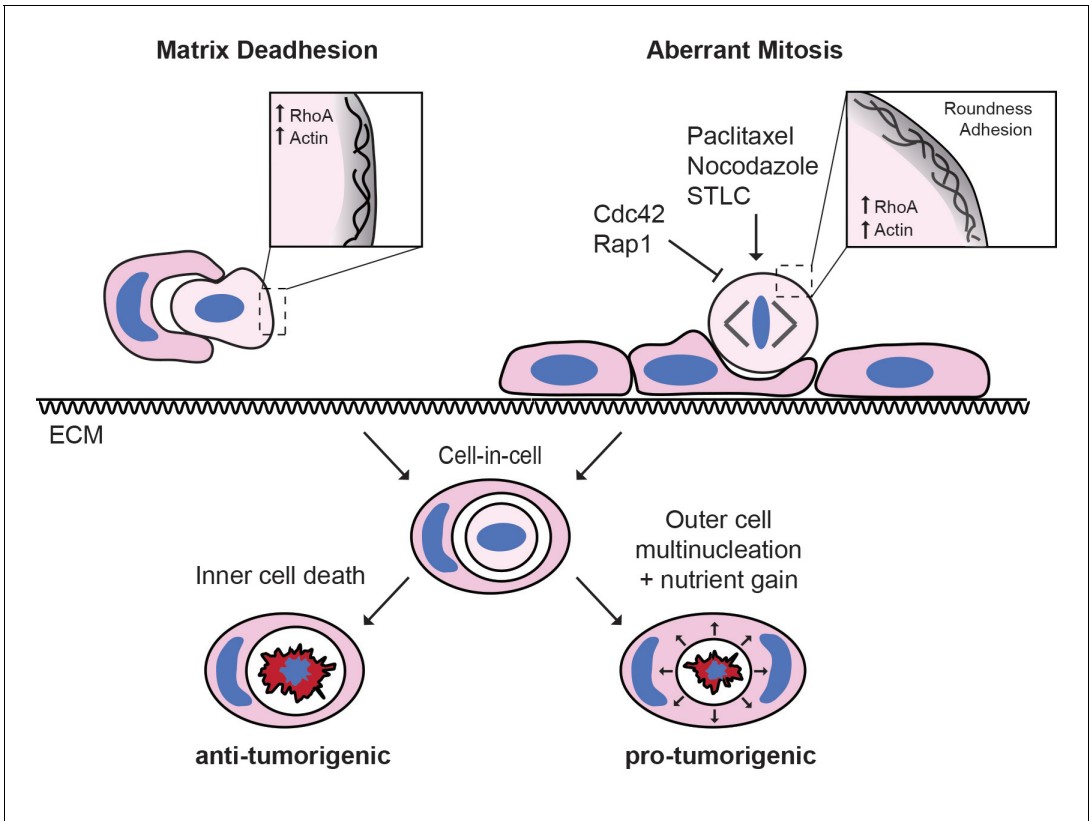

**Figure 8.** The triggers and consequences of entosis in cancer. Entosis can be triggered among epithelial cells through either matrix deadhesion or aberrant mitosis. Mitotic entosis is associated with enhanced deadhesion and rounding during cell division, which can be induced by inhibition of Cdc42 or Rap1, or through prometaphase arrest. RhoA activity is important in both suspension and mitosis-induced entosis, driving ROCK-dependent myosin activation. Regardless of the triggering mechanism, entosis promotes both inner cell death and outer cell nutrient gain and multi-nucleation, with the potential to confer both anti- and pro-tumorigenic effects.

correlates with more aggressive disease (*Krajcovic et al., 2011*) and may provide prognostic indications (*Schwegler et al., 2015*; *Schenker et al., 2017*).

In relation to cancer, we also report the chemotherapeutic induction of mitotic entosis as a novel and unanticipated effect of Paclitaxel/taxol treatment. Interestingly, we find that Paclitaxel enhances mitotic deadhesion and rounding, thereby driving subsequent cell-in-cell formation through entosis. There is a significant debate regarding the mechanism of action of taxane-family drugs (*Weaver, 2014*), which can cause prometaphase arrest, multipolar divisions and cell death (*Zasadil et al., 2014*). As such, these additional and unconventional activities in modifying mitotic morphology and driving cell cannibalism through entosis may be of potential clinical interest.

Finally, we show that the outcome of entosis is broadly conserved, whether it is triggered by deadhesion or mitosis, within an adherent or detached host cell (*Figure 8*). On one hand, entosis drives internalised cell killing, a potential means of limiting growth (*Florey et al., 2011*). We find here that the host cell can be wild-type and fully adherent, developing an intriguing model in which aberrantly dividing cells may penetrate, and be eliminated by, the surrounding normal epithelium, in a tumour-suppressive act of 'assisted suicide'. On the other hand, we also show that mitosis-induced entosis can disrupt host cell division, with the potential to drive genomic instability (*Krajcovic et al., 2011*). In this case, an alternative model emerges in which one aberrant division promotes more, thereby contributing to tumour progression. These models are not mutually exclusive and the overall impact of entosis on tumour biology remains the focus of ongoing work (*Durgan and Florey, 2015*).

In conclusion, we propose that there are at least two mechanistic routes to entosis: loss of matrix-attachment and cell division. It is striking that anchorage independence and unrestrained proliferation, two classic hallmarks of cancer (*Freedman and Shin, 1974*; *Hanahan and Weinberg, 2011*),

can both drive this form of cell cannibalism, so commonly observed in tumours. It will be worthwhile through future work to assess whether additional features of cancer cells (e.g. stemness), or their environments (e.g. hypoxia, nutrient deprivation), may similarly trigger entosis, and to more comprehensively investigate the effects of cell cannibalism on tumour development, progression and evolution.

## Material and methods

### Cell culture and treatments

Cells were obtained from the following sources: 16HBE (*Durgan et al., 2015*) (from lab of Dieter Gruenert, UCSF), MCF7 (*Sun et al., 2014a*) (Lombardi Cancer Center, Georgetown University), HCT116 (*Sun et al., 2014a*) (from lab of David Boone, University of Notre Dame), 293FT (*Durgan et al., 2011*) (Invitrogen); all tested negative for mycoplasma (MSKCC core facility) and were cultured as described previously. The following inhibitors were used: Blebbistatin (100 μM; Sigma, UK), C3 Transferase (1 μg/ml; CT04 Cytoskeleton, Denver, CO), Nocodazole (100 ng/ml; Sigma), RO-3306 (5 μM; Sigma), STLC (20 μM, Santa Cruz, Dalla, TX), Taxol (1 μM; EMD), Y-27632 (10 μM; R&D, Minneapolis, MN); high-grade DMSO was used as a carrier control (1:1000; Sigma).

### Retrovirus production and infection

For stable expression of shRNA (pSUPER, shCdc42.1 and 2, pSiren, shITGB1.1, shITGB1.2) or protein (GFP, RFP-zyxin, HA-DN-Rap1, RhoA biosensor), cells lines were generated by retroviral infection and selection as described previously (*Durgan et al., 2011*). Briefly, cells were seeded at $10^5$ cells/6-well, infected by centrifugation and stable cells were selected with Puromycin (1.5 μg/ml for 16HBE, 2.5 μg/ml for MCF7) for 2–5 days. Short-term stable pools of cells were prepared for each experiment to avoid clonal effects.

### shRNA

shRNA Cdc42 depletions were performed using hairpins cloned into pSUPER Puro vector: shCdc42.1 (cctgatatcctacacaacaaa), shCdc42.2 (cagatgtatttctagtctgtt). $\beta$1 integrin (ITGB1) depletions were performed using hairpins cloned into pSiren Puro vector: shITGB1.1 (gccttgcattactgctgat), shITGB1.2 (gccttgcattactgctgatat). Stable pools were seeded at $2.5 \times 10^4$ cells/24-well and incubated for 3 days before analysis. siRNA depletions were performed using the following duplexes (Dharmacon, Lafayette, CO): siControl (siLamin A/C; D-001620–02), siCdc42.1 (gauuacgaccgcugaguua), siCdc42.2 (cggaauauguaccgacugu), siα-catenin SMARTpool (M-010505-01-0005). 16HBE cells were transfected using Lipofectamine LTX as described previously (*Durgan et al., 2015*). Briefly, cells were seeded at $2.5 \times 10^4$ cells/24-well and transfected with 1.25 μl Lipofectamine LTX + 1.25 μl 20 μM siRNA (or 10 μM + 10 μM for co-depletions) in antibiotic-free media, overnight. Media was changed the following day, with or without inhibitors (see figure legends), and cells incubated for at least 3 days to optimise knockdown levels.

### Western blotting

Western blotting was performed as described previously (*Durgan et al., 2008*). The following antibodies were used in this study: α-catenin (Sigma C2081; RRID:AB_476830; 1:1000; blocked with 5% milk), Cdc42 (BD 610929; RRID:AB_398244; 1:250), GAPDH (Santa Cruz 25778; RRID:AB_10167668; 1:2000), HA (Covance MMS-101R; RRID:AB_291262; 1:2000; blocked with 5% milk), ITGB1 (cytoSM158, kindly provided by Dr Filippo Giancotti; 1:2500 blocked with 5% milk), Rap1 (Millipore 07–916; RRID:AB_2177126; 1:1000; blocked with 5% milk), α-tubulin (Serotec MCA77S; 1:2000; blocked with 5% milk). Representative images of blots are shown.

### FACS

$2 \times 10^5$ shControl or shCdc42-16HBE cells were seeded on 60 mm dishes and incubated for 2 days. These cycling populations were fixed with EtOH and stained with propidium iodide as described previously (*Durgan et al., 2011*). DNA content was analysed using a FACSCaliber to capture 10,000–30,000 events/sample. Analysis was performed using FlowJo software.

## Immunofluorescence

Unless otherwise indicated, immunofluorescence (IF) was performed as described previously (*Durgan et al., 2015*). Briefly, cells were fixed using 3.7% formaldehyde in PBS (10 min, RT), permeablised in 0.5% triton (5 min, RT) and then incubated with primary antibody in PBS (4C, overnight): $\beta$-catenin (BD 610153; RRID:AB_397554; 1:100), p-MLC2 (Cell Signalling 3671L; RRID:AB_330249; 1:100), LC3 (Cell Signalling 4108; RRID:AB_2137703; 1:100), LAMP1 (BD 555798; 1:100), p-Histone H3 (Millipore 06–570; RRID:AB_310177; 1:100), ZO-1 (Invitrogen 61–7300; RRID:AB_2533938; 1:100). Cells were washed in PBS and incubated with Alexa Fluor 488/568 goat anti-mouse/rabbit (H+L) secondary (1:500) and Hoechst 3342 (1µg/ml) for 45 min at RT; where indicated, HCS CellMASK Deep Red (Thermofisher, H32721) or Alexa Fluor 488-phalloidin (Cell Signalling, 8878S) were included, to stain the cell body or actin respectively, according to the manufacturers' guidelines. Cells were washed with PBS, then water, and mounted using Prolong Gold Antifade Mountant (Thermofisher). Image acquisition was performed with a Confocal Zeiss LSM 780 microscope (Carl Zeiss Ltd) equipped with a 40X oil immersion 1.4 NA objective, using Zen software (Carl Zeiss Ltd).

## Live imaging

Cells were seeded on glass-bottomed dishes (Mattek, Ashland, MA) and incubated/treated as shown in figure legends. Where indicated, cells were stained with CellTracker Green CMFDA, CellTracker Red CMTPX or CellMASK deep red plasma membrane stain (Invitrogen, C10046) for 30 min, as recommended by the manufacturer (Thermofisher) and with Hoechst 33342 (1 µg/ml, Sigma), then washed and returned to normal growth media for imaging. All live microscopy were performed in an incubation chamber at 37°C, with 5% $CO_2$; for overnight imaging, media was overlaid with mineral oil to prevent evaporation. For widefield timelapse microscopy, fluorescent and DIC images were acquired every 8 min using a Flash 4.0 v2 sCMOS camera (Hamamatsu, Japan), coupled to a Nikon Ti-E inverted microscope, using a 20 × 0.45 NA objective. Image acquisition and analysis was performed with Elements software (Nikon, Japan). For live confocal imaging, the same microscope, camera and software were used as described in the section above.

## FRET analysis of RhoA biosensor

For emission ratio imaging, 16HBE cells stably expressing the RhoA-FLARE biosensor (a gift from Dr Klaus Hahn, Addgene #12602) were seeded on 35 mm glass bottom dishes (Mattek) and treated with siRNA as indicated. Hoechst 33342 was added prior to image acquisition. FRET images were acquired with a Zeiss 780 confocal system equipped with a 40 × 1.4 NA objective lens. 458 nm excitation light was used to excite the donor (CFP), with donor emission collected, 464–490 nm, and acceptor (YFP) emission collected, 535–588 nm. Images were typically taken with a pixel size of 80 nm, a pixel dwell time of 0.97 ms and the pinhole set to 2 Airy units. Images were processed using ImageJ. Briefly, all images were background subtracted and the FRET image was used to make a binary mask and selection region. The FRET image was then divided by the CFP image yielding a ratio image reflecting RhoA activity. A linear pseudocolour lookup table was applied. Only cells that had a high enough signal-to-noise ratio in both the CFP and FRET signals were used.

## 3D-CLEM

Cdc42-depleted cells were cultured on 35 mm gridded glass-bottomed dishes (MatTek). Cells were stained using CellTracker Green, CellMASK Plasma Membrane dye and Hoechst, then analysed by live confocal microscopy. A mitotic cell-in-cell formation event was identified, imaged and its position recorded. The cells were quickly fixed, processed, imaged and analysed by correlative serial block face scanning electron microscopy as described previously (*Russell et al., 2017*).

## Cell-in-cell formation assays: suspension culture

Cells were trypsinised on day 3 post-seeding or transfection, resuspended in growth media, −/+ inhibitors, and seeded at $10^5$ cells/six-well on ultra-low adhesion dishes (Costar) for 8 hr. Following suspension culture, $5 \times 10^4$ cells were transferred to a glass slide by cytospin at 300 rpm for 3 min, fixed in 10% TCA and analysed by IF/confocal.

## Cell-in-cell formation assays: adherent culture

Cells were seeded on glass coverslips or glass-bottomed dishes, treated −/+ siRNA as indicated, and incubated for 3 days. To analyse the effects of taxol, nocodazole and STLC, the drugs were added to WT cells for a further 24 hr. To analyse the effect of Cdk1 inhibition, a Y-27632 wash-out experiment was performed. Following siRNA transfection (16HBE) or seeding (16HBE-DN-Rap1, MCF7), media was replaced with 10 µM Y-27632, to inhibit entosis. This treatment allowed a mono-layer to form in the absence of cell-in-cell formation, thereby yielding a clean background. 3 days later, Y-27632 was washed out, to permit entosis, and replaced with either control media or 5 µM RO-3306 (a Cdk1 inhibitor). Cells were fixed and analysed 24 hr later.

## Inner/outer cell identity assays

$10^5$ WT or GFP-expressing 16HBE cells were seeded per six-well and transfected with siControl or siCdc42.2, respectively. WT cells were treated with Cell Tracker Red for 30 min, and then both cell lines were trypsinised, mixed in equal proportion, then reseeded at $10^5$ cells/well on 35 mm glass bottomed dishes to yield mixed monolayers. 2 days later, the resulting entotic structures were ana-lysed by live IF/confocal microscopy (d3 post-siRNA). The colours could be reversed with no change in experimental outcome (ie. siControl/GFP cells, siCdc42/WT-red cells).

## Mitotic morphology assays

$10^5$ 16HBE cells were seeded/35 mm glass-bottomed dish and transfected with siCdc42. Three days post-transfection, the cells were stained with CellTracker Green and Hoechst for 30 min, then placed in fresh media for imaging. Metaphase cells were imaged by live DIC and confocal micros-copy, with full z-stacks acquired. Images were analysed using ImageJ software. To measure mitotic cell spreading, cell length was measured in a basal x/y section, cell height in the z, and height/length ratio was plotted. For mitotic rounding, cell shape was analysed in the midplane x/y section of each metaphase cell, scoring for roundness (where 1 = a perfect circle).

## Mitotic spreading assays

$10^5$ 16HBE cells were seeded/35 mm glass-bottomed dish and transfected with siControl or siCdc42. On day 3 post-transfection, Cdc42-depleted cells were stained with Hoechst and treated with C3 (1 µg/ml), Y-27632 (10 µM) or Blebbistatin (100 µM) for a further 4 hr. 10 min prior to imag-ing, each dish of cells was treated with CellMASK deep red plasma membrane stain (Invitrogen, C10046), and cells subjected to live IF/confocal imaging. Metaphase cells were identified by DNA morphology and the basal section imaged. Basal spread area was measured using ImageJ.

## 3D culture

MCF7 cells were seeded in 3D matrigel using established protocols (*Durgan et al., 2011*), and incu-bated to form early stage cysts (2–4 cells) in which matrix contacts are retained, to minimise the induction of detachment-induced entosis. Briefly, four-well, glass-bottomed chamber slides (Lab-Tek II; 155382) were coated with a thin layer of 80% Matrigel Growth Factor Reduced Membrane Matrix (Corning; 356230)/20% Rat Collagen I (Cultrex; 3440-100-01), then overlaid with $5 \times 10^4$ cells in 2% Matrigel/media; 10 µM Y-27632 was included to suppress basal detachment-induced entosis during seeding. Cells were incubated for 24 hr to initiate cyst formation, Y-27632 was then washed out and replaced with fresh media −/+ RO-3306 (5 µM), to allow cell-in-cell formation to proceed in 3D, in the presence or absence of mitosis. Twenty-four hour later, cysts were formalin fixed, stained for actin (488-phalloidin) and DNA (Hoechst) and imaged by confocal microscopy.

## Tissue microarray (TMA)

A human breast cancer microarray was obtained from US Biomax (BR1505b), bearing 75 cases of invasive ductal carcinoma (see Supplementary *Figure 2*). Each case was represented by duplicate formalin-fixed, paraffin-embedded cores, each with a diameter of 1 mm and a thickness of 5 µm. The array was baked at 55°C for 30 min, deparaffinised in xylene (2 × 10 min washes, RT) and rehy-drated through sequential washes (2 × 100% EtOH, 1 × 70% EtOH, 1 × 30% EtOH, 3x H2O; 5 min each at RT). Antigens were retrieved by boiling in 1x citrate buffer (Vector Labs) for 20 min, then cooled back to RT. Blocking (45 min, RT) and antibody incubations were performed in TBS-T/5%

BSA/0.1M Glycine; TBS-T was used for washes. Antibodies and mounting conditions are stated above (see IF), with primaries incubated 4°C/overnight, secondaries at RT/45 min and DAPI at RT/10 min. The array was stained for: (1) $\beta$-catenin, to visualise adherens junctions (where present), (2) p-Histone H3 (S10), a marker of mitosis, and (3) DAPI to visualise nuclear morphology. Each 1 mm core was imaged in whole by DIC and confocal microscopy, using a 40 × 1.4 NA objective and tiling a 6 × 6 grid. For each core, the number of p-Histone H3 (S10)-positive nuclei was counted as an indicator of mitotic activity, and the number of cell-in-cell structures scored as a measure of entosis.

## Xenografts

Six- to 8-week-old female athymic mice were implanted with $\beta-17$ estradiol pellets 3 days before tumor implantation. $1 \times 10^7$ MCF7 cells were injected subcutaneously per tumour, in duplicate. When tumours reached a size of ~150 mm^3, mice were treated with either vehicle, or 15 mg/kg taxol. Twenty-four hour post-treatment, tumours were excised, formalin-fixed and 20 micron sections were prepared (these relatively thick sections permit complete visualisation of whole engulfed cell-in-cell structures). Sections were processed and stained as described for the TMA, and analysed by DIC and IF/confocal ($\beta$-catenin, p-HH3/DNA). To account for tumour heterogeneity, five well-separated fields of view (x/y) were captured from each of three distinct sections per tumour (z; from the top, middle, bottom) using a 40 × 1.4 NA objective. The average number of pHH3-positive cells was scored per FOV, per section. The total number of cell-in-cell structures in each section was scored.

## Statistics

Data were analysed by Mann-Whitney U test and Student t-test as indicated, using Prism 6 software.

## Data availability

All relevant data are available from the authors.

## Acknowledgements

We thank the Babraham Institute Imaging and FACS facilities for technical assistance, Dan Jin for reagents, Adrian Saurin and Nick Ktistakis for helpful comments and Len Stephens for his support. We acknowledge Elisa De Stanchina for xenograft injections at the Antitumor Assessment Core Facility at MSKCC. JD was funded by a Revson Senior Fellowship in the Biomedical Sciences, a Marie-Curie Fellowship (624161) and a L'Oreal for Women in Science UK award, LC by Cancer Research-UK, the MRC, Wellcome Trust, BBSRC and EPSRC (MR/K01580X/1), MO by the National Cancer Institute (RO1CA154649), and OF by a Cancer Research-UK fellowship (CA47718/A16337); we also acknowledge the friends and family of Jean Florey, who donated in her memory. We dedicate this work to our friend and colleague Alan Hall, who so sadly passed away prior to its publication.

## Additional information

### Funding

| Funder | Grant reference number | Author |
|---|---|---|
| Cancer Research UK | CA47718/A16337 | Joanne Durgan Oliver Florey |
| Revson Senior Fellowship | | Joanne Durgan |
| Marie-Curie Fellowship | 624161 | Joanne Durgan |
| L'Oreal & UNESCO UK and Ireland | | Joanne Durgan |
| National Cancer Institute | RO1CA154649 | Jens C Hamann Michael Overholtzer |
| Cancer Research UK | MR/K01580X/1 | Marie-Charlotte Domart Lucy Collinson |
| Medical Research Council | MR/K01580X/1 | Marie-Charlotte Domart Lucy Collinson |

| Wellcome | MR/K01580X/1 | Marie-Charlotte Domart Lucy Collinson |
| Biotechnology and Biological Sciences Research Council | MR/K01580X/1 | Marie-Charlotte Domart Lucy Collinson |
| Engineering and Physical Sciences Research Council | MR/K01580X/1 | Marie-Charlotte Domart Lucy Collinson |

The funders had no role in study design, data collection and interpretation, or the decision to submit the work for publication.

## Author contributions

JD, Conceptualization, Data curation, Formal analysis, Funding acquisition, Investigation, Methodology, Writing—original draft, Project administration, Writing—review and editing; Y-YT, Resources; JCH, Investigation; M-CD, Resources, Investigation; LC, Resources, Funding acquisition, Investigation; AH, Resources, Supervision; MO, Resources, Funding acquisition; OF, Conceptualization, Resources, Data curation, Formal analysis, Supervision, Funding acquisition, Investigation, Methodology, Writing—original draft, Project administration, Writing—review and editing

## Author ORCIDs

Oliver Florey, http://orcid.org/0000-0002-1075-7424

## Ethics

Animal experimentation: Xenograft transplantation studies were performed in full compliance with NIH guidelines and approved with Institutional Animal Care and Use Committee (IACUC) guidelines at Memorial Sloan-Kettering Cancer Center under the approved protocol 04-03-009. Xenograft injections were performed by Elisa De Stanchina at the Antitumor Assessment Core Facility at MSKCC.

## Additional files

### Supplementary files

• Supplementary file 1. Human breast invasive ductal carcinoma tissue microarray. Accompanying tumour information for the Biomax tumour microarray BR1505b used in *Figure 6d*.

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
