## [Decision Letter]

[Editors’ note: a previous version of this study was rejected after peer review, but the authors submitted for reconsideration. The first decision letter after peer review is shown below.]

Thank you for submitting your work entitled "Mitosis can drive cell cannibalism through entosis" for consideration by *eLife*. Your article has been favorably evaluated by Anna Akhmanova (Senior Editor) and three reviewers, one of whom, Α Yap, is a member of our Board of Reviewing Editors. The reviewers have opted to remain anonymous.

Our decision has been reached after consultation between the reviewers. Based on these discussions and the individual reviews below, we regret to inform you that your work will not be considered further for publication in *eLife*.

Let me summarize: we all consider the phenomenon that you report to be interesting and its phenomenology to be convincingly described. However, there are two broad areas where we feel that it is still lacking to be appropriate for *eLife*.

Firstly, we feel that more mechanistic insight is needed. Overall, we think that this is the key area where the manuscript needs to be advanced. You will see that the reviewers have suggested a range of areas that could be investigated, ranging from some measure of the biophysical changes in the to-be-entosed cells (as could be inferred from cell shape changes: Reviewers 1 and 3), to changes in the balance of GTPase signaling and adhesion. Of course, this is a wide range of suggestions, not all of which would need to be followed to reasonably strengthen the manuscript. We will leave it to your judgement to consider what is feasible from your perspective.

Secondly, the link to cancer could also be strengthened. On balance, we think that this is less important. Reviewer 2 has suggested a range of possibilities, some of which (such as the use of organoid or 3D cultures) would be exciting but beyond the reasonable scope of a revision. Here we ask you to focus on 2 discrete aspects: a) What might be the association with prognosis; and b) whether there is a link to a change in GTPase balance (over-activation of RhoA or downregulation of Cdc42).

Our policy is to not ask for extensive revisions that will take more than two months to complete. Therefore, whilst we do find this work interesting, we feel it will take considerably longer than two months to investigate the above points that would make your study a more substantial contribution. Hence, we are rejecting it for now but would be open to a revised manuscript if the mechanistic insights and links to cancer were explored. This paper would be treated as a new submission with no guarantees of acceptance, but we will endeavour to secure the same referees. Alternatively, you are of course also free to submit your work elsewhere if you would like to publish sooner rather than later.

*Reviewer #1:*

This manuscript explores the phenomenon of entosis, a still-enigmatic process whereby cells can be engulfed and ultimately killed, by neighbouring cells. To date, entosis has generally been observed when cells are grown in suspension. In contrast, Durgan et al. now document circumstances when substrate-adherent cells undergo entosis. They show that this is associated with mitosis, when either Cdc42 or Rap1 signaling are perturbed, maneuvers that promote deadhesion and cell rounding. Further, they show evidence that this process may occur in tumors, and can be promoted by chemotherapeutic agents that target microtubules or mitotic kinesins.

Overall, these are interesting observations, but – though thorough – the analysis remains somewhat phenomenological. My appreciation of the (patho)biologic and/or mechanistic significance of these findings would be increased if the authors could address the following questions:

1) Is cortical tension altered in mitotic Cdc42 (or Rap1)-deficient cells? The authors speculate that the fundamental reason why entosis occurs has something to do with the biophysical properties of the Cdc42-deficient mitotic cells. Changes in cortical tension would be an obvious possibility that could beapproached, including direct measurement by AFM or, potentially, inference from the angles between adjacent cell cortices at the division planes.

2) Is the frequency of entosis increased by transformation? The authors provide some evidence to suggest that this may be so, by comparing entosis frequency in tumor specimens that different in density of mitotic cells. However, the relevance of this phenomenon to cancer would be clearer if the authors could compare transformed vs. non-transformed cells.

3) Is the entosis seen in cancer cells causally linked to changes in Cdc42 and/or Rap1 signaling?

*Reviewer #2:*

Durgan and colleagues present data to show that mitosis drives entosis in adherent cultured cells, and that this process requires the cell cycle modulator and Rho GTPase Cdc42. This result is surprising because until now entosis had been thought to occur only in cells that have lost matrix attachment. Furthermore, the authors found that mitotic deadhesion and rounding drives entosis. Consequently, a dominant-negative form of Rap1, a small GTPase that promotes cell spreading, also drives entosis. Lastly, the authors showed that cell-in-cell structures occur in human breast tumors using a tissue array and that taxol treatment drives cell deadhesion and rounding, which promotes entosis – a previously unreported function of the drug.

While the premise of this study is quite interesting, two major issues need resolving to warrant publication.

First, the mechanism for this different mode of entosis is missing and given that this is the main point of this paper, a better understanding of what exactly drives this activity needs elucidation. How does Cdc42 effect changes in the cytoskeleton to drive de-adhesion and rounding? Is this chiefly an outcome of cells that lack cdc42, or a more general activity of cells that are rounded by being Rho-activated? If so, cells that move in Rho-dependent mechanism, such as many cancer cells could also be affected. Therefore, they need to clearly define the base mechanism for this activity.

Second, the relevance of mitotic entosis to human cancer is not clear. Is Cdc42 or mitotic entosis a prognostic indicator in cancer? The relevance of mitotic entosis to the ability of cells to metastasize, escape the immune system, or die would be useful to know, therefore data suggesting its role in cancer would make the story more exciting and compelling. The studies in established cancer cells in culture are not all that convincing, given that the dependence for this activity would not happen on glass. Using organoids made from wild type or cancer tissue would give a more convincing platform to see if mitotic entosis occurs at a higher rate in cancer. Furthermore, the data from the cores is hard to interpret and correlative. From your previous studies, it seems that mitosis can occur after entosis. Were these tumors treated with taxol? Do cancers that stratify with mitotic entosed cells have a poor or good prognosis?

Other major comments

1) What is the reason for looking at Cdc42 in the first place? Are there other cell cycle modulators that phenocopy Cdc42 depletion (in particular, any Cdc42 targets)?

2) Does Cdc42 knockdown affect cell proliferation rate?

3) Do AJs and TJs also break down in MCF7 cells?

4) To show knockdown specificity: does Cdc42 re-expression after knockdown rescue the defect in entosis? Or do siCdc42 cells that are grown longer-term and lose the knockdown display a decrease in cell-in-cell structures?

5) What downstream targets of Cdc42 might be altering the cytoskeleton?

6) Could the Rho-dependent cleavage force of cytokinesis stimulate initiation of entosis? Or does rounding in prometaphase drive it more?

7) The results shown in Figure 6 are baffling and lack proper controls. Does taxol have a pleiotropic effect on cancer cells, i.e., pro-survival by enhancing multinucleation and pro-death by enhancing cell death through entosis (and apoptosis caused by taxol itself)? Instead of showing zoomed in images of cell-in-cell structures, show zoomed out images to indicate the differences in numbers of entotic cells in control vs. drug tx. What is the degree of multinucleation with taxol? Any data in the BC tissue array to indicate taxol tx? Can you tease apart the multiple functions of taxol by co-treating cells with taxol and an entosis inhibitor, or an autophagy or lysosomal inhibitor?

8) Providing a model figure would help solidify the mechanism and take-home message.

*Reviewer #3:*

The authors show that:

Entosis between adherent cells is increased in cdc42-depleted cells. This novel form of entosis bears some characteristics of the one previously described in cells in suspension.

Cdc-42-dependent entosis in adherent cells is associated with and requires mitotic entry.

Cdc42 depleted cells exhibit increased mitotic rounding and deadhesion. Accordingly blockage of GTPase Rap function which affects mitotic rounding and deadhesion promotes entosis in adherent cells.

Loss of matrix-attachment is not sufficient to drive entosis in adherent cells, rather mitotic rounding is the key parameter.

Adherent cell entosis is observed in human carcinomas and cell lines and prevention of mitotic progression using taxane drugs increases entosis.

The reported data fully support the authors' conclusions. Mitotic rounding and deadhesion are essential processes associated with genome stability. Overall the reported findings are of general interest.

1) The authors show that adherent entosis requires α-catenin. Are cdc42- and α-catenin depleted cell less round or more adhesive during mitosis? Similarly, could the authors describe the shape of Rocki and cdc42 depleted cells? The role of α-catenin and Rock in adherent cell entosis should be at least discussed in the light of the novel findings of the authors.

2) The authors exclude that entosis is triggered by spindle mis-orientation since the depletion of Par-3 or aPKC lead to spindle mis-orientation without observable increase of entosis. The authors should at least report the percentage of entosis in the par-3 depletion cells.

[Editors’ note: what now follows is the decision letter after the authors submitted for further consideration.]

Thank you for submitting your article "Mitosis can drive cell cannibalism through entosis" for consideration by *eLife*. Your article has been favorably evaluated by Anna Akhmanova (Senior Editor) and three reviewers, one of whom is a member of our Board of Reviewing Editors. The reviewers have opted to remain anonymous.

The reviewers have discussed the reviews with one another and the Reviewing Editor has drafted this decision to help you prepare a revised submission.

Summary:

While we have treated it formally as a new submission, it has been seen by the original reviews. As you can see from their comments, they appreciate that you have made substantial efforts to address the issues raised in their earlier reviews and are supportive of publication.

Essential revisions:

Reviewers 2 and 3 raise some residual issues for revision that look as if they can be addressed with minor re-writing and some re-analysis of the data. Reviewer 3 also identified some interesting potential experiments that could pursue the role of RhoA signaling, but, after consultation, we agree that these could well form the basis for a follow-up study, rather than being essential for your present manuscript.

*Reviewer #1:*

Overall, the authors have responded reasonably to the issues that I raised in my earlier review of their manuscript. I appreciate the analytical challenges that would be entailed by more detailed e.g. biophysical analysis of this interesting phenomenon. The experiments that they have added expand and clarify the manuscript. Therefore, I think that this is a valuable manuscript which opens up avenues for future research and support its publication in *eLife*.

*Reviewer #2:*

Durgan and colleagues present data to show that mitotic deadhesion and rounding drives entosis in adherent cultured cells upon Cdc42 depletion. This is a surprising finding because until now, entosis had been thought to occur only in cells in suspension. They also show that this process depends on cell rounding through Rho-, ROCK-, and myosin-dependent, and that dominant-negative Rap1 drives mitotic deadhesion and, consequently, entosis. Lastly, the authors showed that cell-in-cell structures are formed in cancer cell lines and patient breast tumor samples, and that taxol treatment promotes entosis both in cultured cells and in mouse xenografts – a previously unreported function of the drug.

All in all, this is a much improved manuscript and the writing is far more compelling and clearer. The authors addressed concerns about the mechanistic detail of this process sufficiently.

1) Showing that β1-integrin is necessary (subsection “Enhanced mitotic deadhesion and rounding can induce entosis: Rap1”) for entosis here is necessary, as Cdc42 has been shown previously to regulate β1-integrin.

2) For some, it may be confusing about mitotic blockers, as Cdk1 inhibitor blocks entosis and taxol promotes it. Does the Cdk1 inhibitor block mitotic deadhesion and rounding, whereas the other drugs promote it by arresting cells in mitosis?

3) Regarding the relevance of Cdc42 to cancer, could the authors comment on whether Cdc42 expression levels have been shown to impact metastases and/or patient survival? Do MCF7 and HCT116 cells perhaps express reduced levels of Cdc42, compared to normal cells (i.e. MCF10A)? Or is this just a link to cell roundedness independent of cdc42 by increased Rho activity. The answer doesn't really matter here but it just needs more clarity as to why they went from the results on rounding to cancer situations.

*Reviewer #3:*

The authors have addressed my comments in detail. I note that they have not demonstrated that rhoA activation is necessary for entosis in the cdc42 knock-down conditions. The authors mentioned that "Unfortunately, it is not possible to demonstrate this link unambiguously, because RhoA inhibition will block entotic cell-in-cell formation regardless of its trigger, due to downstream effects on myosin contractility (Overholtzer et al., 2007) and actin dynamics (Purvanov et al., 2014)".

Do they mean that RhoA is required in the engulfed or the engulfing cells?

Have the authors considered to address the role of RhoA dependent contractility in adherent cell entosis by:

Using a RhoA-depleted and wt mixed cell population as performed in Figure 2 for cdc42-depleted versus wt cells?

Analysing other regulators of the cell cortex contractility during mitosis such as ERM proteins?

---

## [Author Response]

[Editors’ note: the author responses to the first round of peer review follow.]

*Reviewer #1:*

*[…] Overall, these are interesting observations, but – though thorough – the analysis remains somewhat phenomenological. My appreciation of the (patho)biologic and/or mechanistic significance of these findings would be increased if the authors could address the following questions:*

*1) Is cortical tension altered in mitotic Cdc42 (or Rap1)-deficient cells? The authors speculate that the fundamental reason why entosis occurs has something to do with the biophysical properties of the Cdc42-deficient mitotic cells. Changes in cortical tension would be an obvious possibility that could beapproached, including direct measurement by AFM or, potentially, inference from the angles between adjacent cell cortices at the division planes.*

We agree with the reviewer’s suggestion that cortical changes during cell division seem a likely mechanism during mitotic entosis. Although we considered measuring tension directly using AFM, we were concerned about technical issues in this particular system. Given the nature of the Cdc42 phenotype, our comparison would have to be made between control cells, which are adherent during division, and Cdc42-knockdowns, which are largely de-adhered in mitosis. Unfortunately, AFM is not readily amenable to the analysis of loosely adherent cells, which tend to slide in response to the cantilever. Given these methodological issues, we sought an alternative approach, and focussed on an analysis of molecular events at the mitotic cortex, specifically the spatiotemporal regulation of RhoA activity and actin organisation,

both of which are known to be associated with rigidity and rounding during cell division^3^.

These experiments were undertaken using a FRET-based biosensor to localise and monitor RhoA activity through time, and by phalloidin staining of actin in fixed cells, as shown in the new Figure 4. We find that Cdc42-depletion leads to overactivation of RhoA at the mitotic cortex, and observe a corresponding enrichment of cortical actin during metaphase, both consistent with altered rigidity. These data provide important new insight into the molecular mechanisms underlying mitotic entosis, which we have developed further with additional, inhibitor-based studies (See reviewer 2, response 1).

*2) Is the frequency of entosis increased by transformation? The authors provide some evidence to suggest that this may be so, by comparing entosis frequency in tumor specimens that different in density of mitotic cells. However, the relevance of this phenomenon to cancer would be clearer if the authors could compare transformed vs. non-transformed cells.*

In response to this comment, we can note that when working with immortalised, but non-transformed cell lines, such as 16HBE and MCF10A, we very rarely observe mitotic entosis under basal conditions, but instead induce this phenotype through Cdc42 depletion, Rap1 inhibition, or drug treatment. In contrast, in certain transformed cell lines, such as MCF7, Hct116 and MCAS, constitutive mitosis-induced entosis is readily detected under basal conditions, in both 2D and 3D culture. These observations are consistent with the notion that mitotic entosis may be more prevalent among transformed cells, a hypothesis that is more convincingly evidenced by our data in primary human tumour samples.

*3) Is the entosis seen in cancer cells causally linked to changes in Cdc42 and/or Rap1 signaling?*

Our data indicate that mitotic entosis can be induced by suppression of Cdc42 or Rap1 signalling, but also by a variety of drugs that induce prometaphase arrest and enhance mitotic rounding. In light of these data, we reason that this phenotype is unlikely to be specific to the Cdc42/Rap1 pathways, which have not been widely linked to cancer, but rather occurs more broadly in response to genetic changes and/or drug treatments that promote alterations in mitotic morphology.

*Reviewer #2:*

*[…] While the premise of this study is quite interesting, two major issues need resolving to warrant publication.*

*First, the mechanism for this different mode of entosis is missing and given that this is the main point of this paper, a better understanding of what exactly drives this activity needs elucidation. How does Cdc42 effect changes in the cytoskeleton to drive de-adhesion and rounding? Is this chiefly an outcome of cells that lack cdc42, or a more general activity of cells that are rounded by being Rho-activated? If so, cells that move in Rho-dependent mechanism, such as many cancer cells could also be affected. Therefore, they need to clearly define the base mechanism for this activity.*

Reviewer 2 suggests that a better understanding of the molecular mechanisms underlying mitotic entosis is required, and more specifically, an analysis of the contribution made by RhoA. To address this important point, we have undertaken a FRET-based analysis of spatiotemporal RhoA signalling, in the presence and absence of Cdc42, as outlined above (reviewer 1, response 1). These experiments indicate that loss of Cdc42 permits overactivation of cortical RhoA during mitosis, which is accompanied by an enrichment of actin at the metaphase cortex (Figure 4). Building on these findings, we went on to show that the effects of Cdc42 depletion can be reverted by inhibition RhoA, or its downstream effectors, ROCK and myosin (Figure 4). Together, these data develop a model in which loss of Cdc42 permits the overactivation of a RhoA/ROCK/myosin cascade, which drives enhanced mitotic deadhesion and rounding and subsequent entosis. Notably, we find that loss of adhesion during interphase is insufficient to drive entotic penetration of an adherent host cell (Figure 5). Similarly, we have found that interphase activation of myosin phosphorylation, through depletion of MYPT1, is also insufficient to drive adherent entosis (data not shown). As such, we believe this Cdc42-regulated RhoA/ROCK/myosin cascade, and its effects on entosis, are intimately linked to mitosis.

*Second, the relevance of mitotic entosis to human cancer is not clear. Is Cdc42 or mitotic entosis a prognostic indicator in cancer? The relevance of mitotic entosis to the ability of cells to metastasize, escape the immune system, or die would be useful to know, therefore data suggesting its role in cancer would make the story more exciting and compelling.*

These are all interesting suggestions. However, in fixed patient samples, mitotic entosis cannot be distinguished from cell-in-cell formation triggered through other mechanisms. As such, it is not possible to focus specifically on the correlation between mitotic entosis and prognosis (or metastasis, immune evasion or cell death). A more general study could be conceived, in which cell-in-cell structures, formed through any mechanism, are correlated with this broad range of cancer features. However, this would not

seem directly relevant to the current work. We draw your attention to emerging research that has explored the prognostic value of cell-in-cell formation in cancer^1,2^, and agree this is an area that warrants further work in the future.

*The studies in established cancer cells in culture are not all that convincing, given that the dependence for this activity would not happen on glass. Using organoids made from wild type or cancer tissue would give a more convincing platform to see if mitotic entosis occurs at a higher rate in cancer.*

Although we appreciate the advantages of organoid models, and have exploited them ourselves in other studies^4^, the issue of matrix binding somewhat complicates their use in this particular study. In most 3D-cyst based models, the central cells are matrix-deprived and therefore may be prone to undergo entosis through deadhesion, complicating our analyses. In light of this issue, we do not believe mature cysts are well suited to the study of mitotic entosis. Nevertheless, we accept the concern over cell culture on glass and have therefore analysed MCF7 breast cancer cells, cultured in 3D matrigel but imaged very early in cyst development (2-4 cell stage), at a point when all cells retain an interface with the matrix. These findings, presented in Figure 6, provide evidence for mitotic entosis in 3D, as well as 2D, culture. We also draw your attention to Figure 7, which demonstrates that MCF7 cells undergo increased entosis upon mitotic arrest in mouse xenografts, an in vivo 3D context. We contend that together, these data address the concern of physiological relevance.

*Furthermore, the data from the cores is hard to interpret and correlative. From your previous studies, it seems that mitosis can occur after entosis. Were these tumors treated with taxol? Do cancers that stratify with mitotic entosed cells have a poor or good prognosis?*

While we acknowledge that the tumour microarray data are correlative, we would argue that this is the best experiment that can currently be performed to address the occurrence of mitotic entosis in human cancer, and furthermore, that the data agree well with our more comprehensive observations in cell lines. We test our model further by manipulating mitotic index in mouse xenografts using Paclitaxel/taxol, and observe an increase in cell-in-cell formation that is also consistent with mitotic entosis among tumour cells in vivo. By combining all of these data, we believe we have built a convincing body of evidence to support the notion that mitotic entosis occurs within tumours.

The tumour microarrays used were obtained from Biomax. We have confirmed with the company that the cores come from untreated patients, so Paclitaxel/taxol is not relevant here. We do not have access to further clinical data related to these particular samples, but refer to reviewer 2, response 2 in relation to the issue of prognostics.

*Other major comments*

*1) What is the reason for looking at Cdc42 in the first place? Are there other cell cycle modulators that phenocopy Cdc42 depletion (in particular, any Cdc42 targets)?*

As outlined in the paper (see Introduction and text accompanying Figure 1), this study was initiated to investigate a possible role for Cdc42 in cell-in-cell formation among detached cells. Cdc42 is known to control epithelial junction formation, cytoskeletal organisation and myosin contractility, all of which might be expected to influence entosis, and thus it was somewhat surprising that Cdc42 depletion had very little effect on entosis among suspension cells. The observation that Cdc42 depletion could induce cell-in-cell formation among adherent cells was entirely unexpected, but provided valuable new insights, opening up the discovery of mitotic entosis.

As described in the Results, we have phenocopied loss of Cdc42 by inhibiting Rap1, a known regulator of mitotic spreading, and by modulating cell cycle progression with spindle poisons (taxol, nocodazole) and a mitotic kinesin inhibitor (STLC). All of these modulators similarly enhance mitotic deadhesion and rounding, and induce subsequent mitotic entosis.

*2) Does Cdc42 knockdown affect cell proliferation rate?*

As shown in Figure 3, there is no dramatic change in cell cycle progression following Cdc42 depletion.

*3) Do AJs and TJs also break down in MCF7 cells?*

It is not entirely clear what the reviewer is asking here, but we assume the question relates to MCF7 junctions in the presence or absence of Cdc42? If so, we can confirm that AJs and TJs are intact in wild-type MCF7s, but less mature in Cdc42-depleted, consistent with findings in 16HBE.

*4) To show knockdown specificity: does Cdc42 re-expression after knockdown rescue the defect in entosis? Or do siCdc42 cells that are grown longer-term and lose the knockdown display a decrease in cell-in-cell structures?*

We consider that the use of 4 distinct and non-overlapping RNAi reagents, of different types (2 siRNA duplexes and 2 shRNA hairpins), provides sufficiently robust evidence for the specificity of this phenotype, particularly as the level of Cdc42-knockdown is shown clearly to correlate with the strength of phenotype (see siCdc42.1 v siCdc42.2; Figure 1 and accompanying text). We note that we have observed the same phenotype with an additional Cdc42-specific shRNA (data not shown) and confirm that we have never come across an si- or shRNA reagent which knocks down Cdc42, but does not give this phenotype. We contend that we have already provided adequate evidence on this point.

We have not set up an RNAi-rescue for Cdc42, because it has been reported previously that even modest overexpression of this gene itself yields phenotypic changes, confounding interpretation^5^.

We can confirm that prolonged passage of the cells does lead to a gradual loss of both knockdown and phenotype. However, we do not find this a very convincing proof of specificity, as this could presumably be the case regardless of whether the effect was on- or off-target.

*5) What downstream targets of Cdc42 might be altering the cytoskeleton?*

This is an interesting question. To address this issue, we conducted a SMARTpool-based siRNA screen of Rho-family effectors, assaying for the induction of adherent entosis.

Although this screen yielded 3 preliminary hits, none were robustly validated through follow up work with additional siRNAs, and therefore seem likely to have been off-target. Our inability to identify a single Cdc42 target may be a limitation of the library used, as we cannot be sure that all SMARTpools work efficiently, or may perhaps indicate redundancy, as some of the targets are grouped into families (e.g. PAKs 1-3 and 4-6). More interestingly, as mitotic entosis is a multi-step process, it may represent a more complex phenotype in which multiple hits are required simultaneously. Although interesting, we feel that further analysis of this point would go beyond the scope of this paper.

*6) Could the Rho-dependent cleavage force of cytokinesis stimulate initiation of entosis? Or does rounding in prometaphase drive it more?*

This is an interesting point, and our new data (Figure 4) certainly implicate RhoA activity in both mitotic rounding and entosis. Based on timelapse imaging, we have observed cell-in-cell formation occurring at various stages of mitosis. In some cases, and of course particularly upon taxol/nocodazole/STLC treatment, entosis occurs during prometaphase. In other cases, one or both daughters are internalised during/shortly after cytokinesis. While the general biophysical changes associated with mitosis are clearly important for adherent entosis, we have not found evidence of an obvious link with one particular phase of mitosis compared to another.

*7) The results shown in Figure 6 are baffling and lack proper controls.*

It is not clear exactly what the reviewer finds confusing, or what controls they consider to be lacking? The rationale for this experiment was to test the model of mitotic entosis in cell lines and in vivo, by manipulating mitotic index with drugs, and then assessing the effect on cell- in-cell formation. We clearly demonstrate that taxol, nocodazole and STLC all increase mitotic index and induce a corresponding increase entosis, consistent with the process of mitotic entosis. Furthermore, these studies reveal that these drugs also significantly enhance mitotic deadhesion and rounding, providing independent support for a link between mitotic morphology changes and cell-in-cell formation. We feel these experiments are well designed and the data clear. These findings are also important with respect to identifying a novel activity of the commonly used chemotherapeutic drug Paclitaxel.

*Does taxol have a pleiotropic effect on cancer cells, i.e., pro-survival by enhancing multinucleation and pro-death by enhancing cell death through entosis (and apoptosis caused by taxol itself)?*

As our goal was to assay for the occurrence of mitotic entosis, our Paclitaxel/taxol experiments were timed accordingly to optimise this, with samples analysed at 24hrs post- treatment in both cell culture and mice. At this early timepoint, the occurrence of mitotic entosis is readily detectable, but there is little opportunity to analyse subsequent effects on cancer cell survival or death. Conceptually, it seems unlikely that multinucleation would be observed in the context of taxol, as the host cell would be prone to arrest in prometaphase too. However, we would expect the effects of taxol to be pleiotropic, as this drug is well known to cause multi-polar division and cell death, as well as prometaphase arrest^6^.

Through our work, we can now add mitotic rounding and cell cannibalism to the list of effects Paclitaxel/taxol can induce, which we believe will be a useful new insight to share with the field.

*Instead of showing zoomed in images of cell-in-cell structures, show zoomed out images to indicate the differences in numbers of entotic cells in control vs. drug tx.*

A zoomed in image is presented in Figure 7 to demonstrate the appearance of a typical cell-in-cell structure within a xenograft sample, as clearly as possible. A zoomed out image would not be particularly helpful in this case, as these structures will not be readily distinguishable (unlike mitotic cells, which are easy to recognise as bright pHH3+ve spots, even at low resolution, Figure 7). Given this, the bar graph presented in Figure 7, which indicates scoring from multiple slices across multiple xenografts, provides much more meaningful quantitative data than a single image.

*What is the degree of multinucleation with taxol?*

The timing of our experiment is not compatible with this analysis. However, the expectation would be that since taxol causes mitotic arrest, multinucleation is an unlikely outcome.

*Any data in the BC tissue array to indicate taxol tx?*

According to Biomax, the company who supplied the TMA, the patients are all untreated.

*Can you tease apart the multiple functions of taxol by co-treating cells with taxol and an entosis inhibitor, or an autophagy or lysosomal inhibitor?*

We have not attempted to explore whether mitotic entosis makes a functional impact on taxol treatment. We feel the suggested experiments go beyond the scope of this study and would not be specific enough to draw strong conclusions. The only inhibitor that has been used to block entosis in vivo is Y-27632. This drug will block all functions of ROCK (and PKN), and thus cannot be used to implicate entosis specifically. With regards to autophagy and lysosomal inhibitors, we assume the reviewer is thinking of a method with which to manipulate entotic corpse degradation? However, these drugs will also have a range of other effects and, again, cannot be used to dissect the effects of entosis specifically.

*8) Providing a model figure would help solidify the mechanism and take-home message.*

Thanks for this suggestion, we have added a new Figure 8 to clarify our main conclusions.

*Reviewer #3:*

*[…] 1) The authors show that adherent entosis requires α-catenin. Are cdc42- and α-catenin depleted cell less round or more adhesive during mitosis? Similarly, could the authors describe the shape of Rocki and cdc42 depleted cells? The role of α-catenin and Rock in adherent cell entosis should be at least discussed in the light of the novel findings of the authors.*

These are interesting suggestions, and analysis of ROCK in particular has proven important for our work; the data are included in the new Figure 4. As noted above (reviewer 2, response 1), we find that inhibition of ROCK (Y-27632), RhoA (C3) or myosin (Blebbistatin), reverts the retraction phenotype induced by Cdc42-depletion during mitosis. Combined with our FRET-based analysis of RhoA activity, these data suggest that Cdc42 constrains the cortical activation of RhoA during epithelial cell mitosis. Loss of Cdc42 permits overactivation of a RhoA/ROCK/myosin cascade, which can drive mitotic rounding and entosis.

With regards to α-catenin, we have not observed an obvious difference in mitotic morphology between cells depleted of Cdc42 alone and those co-depleted of both Cdc42 and a-catenin; all show reduced spreading and increased rounding. We assume that the defect in adherent entosis observed upon a-catenin depletion is instead related to the cells’ inability to form the cell-cell contacts which initiate subsequent internalisation.

*2) The authors exclude that entosis is triggered by spindle mis-orientation since the depletion of Par-3 or aPKC lead to spindle mis-orientation without observable increase of entosis. The authors should at least report the percentage of entosis in the par-3 depletion cells.*

In the original text, we had depleted Par6B (not Par3) and aPKC to explore the possible role of the Cdc42-dependent polarity complex during mitotic entosis. As noted by the reviewer, loss of Par6B/aPKC did not phenocopy Cdc42-depletion in this context, even though these proteins so often function together. This finding helps to exclude certain mechanisms as being sufficient to trigger mitotic entosis, such as disruption of junction formation, spindle orientation or control of the metaphase cortex.

We take the point that this data set was not reported thoroughly in our previous draft and have focussed on aPKC here to provide more comprehensive and quantitative findings. In the new Figure 3—figure supplement 1, we first verify aPKC knockdown, using a duplex that codepletes aPKCɩ and aPKCζ and an antibody that recognises both isoforms. We then quantify the associated defect in junction formation, to confirm a functional loss of the protein(s), and as requested, show graphically that there is no detectable entosis observed among aPKC-depleted cells under adherent conditions, across multiple independent experiments.

Our data suggest that defects in aPKC-dependent phenotypes, such as junction formation and spindle orientation, are not sufficient to drive mitotic entosis. However, we do note that we cannot completely exclude the possibility that aPKC may play some role during mitotic entosis. It remains formally possible that a stronger knockdown/knockout, or codepletion with another pathway, may manifest an effect. Thus, to be cautious, we use these data to exclude certain cellular mechanisms, rather than the aPKC gene itself.

References

1) Schwegler, M. et al. Prognostic Value of Homotypic Cell Internalization by Nonprofessional Phagocytic Cancer Cells. Biomed Res Int 2015, 359392–14 (2015).

2) Schenker, H., Büttner-Herold, M., Fietkau, R. & Distel, L. V. Cell-in-cell structures are more potent predictors of outcome than senescence or apoptosis in head and neck squamous cell carcinomas. Radiat Oncol 12, 21 (2017).

3) Matthews, H. K. et al. Changes in Ect2 localization couple actomyosin-dependent cell shape changes to mitotic progression. Dev. Cell 23, 371–383 (2012).

4) Durgan, J., Kaji, N., Jin, D. & Hall, A. Par6B and atypical PKC regulate mitotic spindle orientation during epithelial morphogenesis. J. Biol. Chem. 286, 12461–12474 (2011).

5) Wallace, S. W., Durgan, J., Jin, D. & Hall, A. Cdc42 regulates apical junction formation in human bronchial epithelial cells through PAK4 and Par6B. Mol. Biol. Cell 21, 2996–3006 (2010).

6) Zasadil, L. M. et al. Cytotoxicity of paclitaxel in breast cancer is due to chromosome missegregation on multipolar spindles. Sci Transl Med 6, 229ra43–229ra43 (2014).

[Editors' note: the author responses to the re-review follow.]

*Reviewer #2:*

*[…] 1) Showing that β1-integrin is necessary (subsection “Enhanced mitotic deadhesion and rounding can induce entosis: Rap1”) for entosis here is necessary, as Cdc42 has been shown previously to regulate β1-integrin.*

Thank you for highlighting this omission. Our data indicate that loss of b1-integrin is insufficient to induce entosis among adherent cells and we have included a new Supplementary Figure (Figure 5—figure supplement 1) to show this, accompanied by text in the subsection “Enhanced mitotic deadhesion and rounding can induce entosis: Rap1”.

Please note: we assume that there was a typo in the reviewer 2, point 1 text, as we had previously indicated (and now show) that b1-integrin is unnecessary, rather than necessary, for entosis under these conditions.

*2) For some, it may be confusing about mitotic blockers, as Cdk1 inhibitor blocks entosis and taxol promotes it. Does the Cdk1 inhibitor block mitotic deadhesion and rounding, whereas the other drugs promote it by arresting cells in mitosis?*

Thank you for drawing our attention to this potential for confusion. We believe the key difference between Cdk1i versus taxol/nocodazole/STLC in this context lies with the timing of cell cycle arrest. Cdk1 inhibition arrests cells at the G2/M transition, thereby inhibiting mitotic deadhesion and rounding as well as consequent entosis. Taxol, nocodazole and STLC, on the other hand, arrest cells in prometaphase, during which they do deadhere and round; indeed all 3 of these drugs actually enhance these morphological changes. Consistent with our model of mitotic entosis, these drugs also promote associated cell-in-cell formation. We have added text to clarify the difference between these drug induced effects in the subsection “Taxane treatment promotes mitosis-induced entosis”.

*3) Regarding the relevance of Cdc42 to cancer, could the authors comment on whether Cdc42 expression levels have been shown to impact metastases and/or patient survival? Do MCF7 and HCT116 cells perhaps express reduced levels of Cdc42, compared to normal cells (i.e. MCF10A)? Or is this just a link to cell roundedness independent of cdc42 by increased Rho activity. The answer doesn't really matter here but it just needs more clarity as to why they went from the results on rounding to cancer situations.*

This is an interesting point. Although it is tempting to speculate that Cdc42 deregulation could be directly involved in tumour cell cannibalism, this specific GTPase has not been widely implicated in cancer and so we have not tested for this directly. We would consider it more likely that there are multiple pathways that can enhance mitotic cell rounding to drive entosis, perhaps converging on activation of RhoA, ROCK and myosin.

Our rationale for exploring cancer cell lines was that a) cancer cells tend to undergo deregulated divisions, and b) mitotic rounding has emerged as a process that may be relevant in cancer biology (e.g. Cadart et al., 2014). We have attempted to outline this reasoning more clearly in the subsection “Mitosis-induced entosis occurs constitutively in adherent cancer cell lines and human 258 tumours with pleiotropic effects”.

*Reviewer #3:*

*The authors have addressed my comments in detail. I note that they have not demonstrated that rhoA activation is necessary for entosis in the cdc42 knock-down conditions. The authors mentioned that "Unfortunately, it is not possible to demonstrate this link unambiguously, because RhoA inhibition will block entotic cell-in-cell formation regardless of its trigger, due to downstream effects on myosin contractility* (*Overholtzer et al., 2007) and actin dynamics (Purvanov et al., 2014)".*

*Do they mean that RhoA is required in the engulfed or the engulfing cells?*

In the context of detachment-induced entosis, it is well established that RhoA activity in the internalising cell promotes contractility and stiffening through ROCK and myosin, to facilitate the ‘invasion’ of a more physically deformable host^1,2^. Our data suggest a similar pathway operates during mitotic entosis, which is also dependent on ROCK activation and similarly involves localised myosin phosphorylation the internalising cell tail.

*Have the authors considered to address the role of RhoA dependent contractility in adherent cell entosis by:*

*Using a RhoA-depleted and wt mixed cell population as performed in Figure 2 for cdc42-depleted versus wt cells?*

*Analysing other regulators of the cell cortex contractility during mitosis such as ERM proteins?*

We agree that these are interesting questions that, as noted by the editor, could form the basis of follow up work.

References

1) Overholtzer, M. et al. A nonapoptotic cell death process, entosis, that occurs by cell-in-cell invasion. Cell 131, 966–979 (2007).

2) Sun, Q. et al. Competition between human cells by entosis. Cell Res. 24, 1299–1310 (2014).